# LEARNING GRAPH REPRESENTATIONS FOR INFLUENCE MAXIMIZATION

## ABSTRACT

As the field of machine learning for combinatorial optimization advances, traditional problems are resurfaced and readdressed through this new perspective. The overwhelming majority of the literature focuses on small graph problems, while several real-world problems are devoted to large graphs. Here, we focus on two such problems: influence estimation, a #P-hard counting problem, and influence maximization, an NP-hard problem. We develop GLIE, a Graph Neural Network (GNN) that inherently parameterizes an upper bound of influence estimation and train it on small simulated graphs. Experiments show that GLIE provides accurate influence estimation for real graphs up to 10 times larger than the train set. More importantly, it can be used for influence maximization on considerably larger graphs, as the predictions ranking is not effected by the drop of accuracy. We develop a version of Cost Effective Lazy Forward optimization with GLIE instead of simulated influence estimation, surpassing the benchmark for influence maximization, although with a computational overhead. To balance the time complexity and quality of influence, we propose two different approaches. The first is a Q-network that learns to choose seeds sequentially using GLIE's predictions. The second defines a provably submodular function based on GLIE's representations to rank nodes fast while building the seed set. The latter provides the best combination of time efficiency and influence spread, outperforming SOTA benchmarks.

## 1 INTRODUCTION

Several real-world problems can be cast as a combinatorial optimization problem over a graph. From distributing packages (Mathew et al., 2015) to improving the general health (Wilder et al., 2018) and vehicles' management (Touati-Moungla & Jost, 2012), optimization on graphs lies in the core of many real-world applications that are vital to our way of living. Unfortunately, the majority of these problems are NP-hard, and hence we can only approximate their solution in a satisfactory time limit that matches the real world requirements. Recent machine learning methods have emerged as a promising solution to develop heuristic methods that provide fast and accurate approximations (Bengio et al., 2020). The general idea is to train a supervised or unsupervised learning model to infer the solution given an unseen graph and the problem constraints. The models tend to consist of Graph Neural Networks (GNNs) to encode the graph and the nodes, Q-learning (Watkins & Dayan, 1992; Sutton & Barto, 2018) to produce sequential predictions, or a combination of both. The practical motivation behind learning to solve combinatorial optimization problems, is that inference time is faster than running an exact combinatorial solver (Joshi et al., 2019). That said, specialized combinatorial algorithms like CONCORDE for the *Traveling Salesman Problem* (TSP) or GUROBI in general, cannot be surpassed yet (Kool et al., 2018).

Though many such methods have been proposed for a plethora of problems, influence maximization (IM) has not been addressed yet extensively. IM addresses the problem of finding the set of nodes in a network that would maximize the number of nodes reached by starting a diffusion from them (Kempe et al., 2003). The problem is proved to be NP-hard, from a reduction to the set-cover problem. Moreover, the influence estimation (IE) problem that is embedded in IM, i.e., estimating the number of nodes influenced by a given seed set, is #P-hard as it is analogous to counting s-t connectedness and would require $2^{|E|}$ possible combinations to compute exactly, where $|E|$ is the number of network edges (Wang et al., 2012). Typically, influence estimation is approximated using

repetitive Monte-Carlo (MC) simulations of the independent cascade (IC) diffusion model. In general, the seed set is built greedily, taking advantage of the submodularity of the influence function which guarantees an at least $(1 - \frac{1}{e})$ approximation to the optimal. Although the latter lacks of efficiency as one still has to estimate influence for every candidate seed in every step of building the seed sets. Hence, several scalable algorithms Borgs et al. (2014); Tang et al. (2015) and heuristics Chen et al. (2009); Jung et al. (2012) were developed capitalizing on sketches or the structure of the graph to produce more efficient solutions. IM can be applied on a plethora of real-world tasks, such as epidemic containment (Wilder et al., 2017), diminishing fake news (Budak et al., 2011), and running viral marketing campaigns (Domingos & Richardson, 2001).

We address IM using neural networks, to capitalize on the aforementioned advantages as well as their ability to easily incorporate contextual information such as user profiles and topics (Tian et al., 2020), a task that remains unsolvable for non-specialized IM algorithms and heuristics. We propose GLIE, a GNN that provides efficient IE for a given seed set and a graph with influence probabilities. It can be used as a standalone influence predictor with competitive results for graphs up to 10 times larger than the train set. Moreover, we leverage GLIE for IM, combining it with CELF (Leskovec et al., 2007), that typically does not scale beyond networks with thousands of edges. The proposed method runs in networks with millions of edges in seconds, and exhibits better influence spread than a state-of-the-art algorithm and previous GNN-RL methods for IM. In addition, we develop GRIM, a Q-learning architecture that utilizes GLIE's representations and predictions to obtain seeds sequentially, while minimizing the number of influence estimations throughout steps. Finally, we propose PUN, a method that uses GLIE's representations to compute the number of neighbors predicted to be uninfluenced and uses it as an approximation to the marginal gain. We prove PUN's influence spread is submodular and monotone, and hence can be optimized greedily with a guarantee, in contrast to prior learning-based methods. The experiments indicate that PUN provides the best balance between influence quality and computational efficiency.

The paper is organized as follows. Section 2 presents an overview of relevant approaches and clarifies the advantage of the proposed models. Section 3 describes the proposed methods, starting with IE and advancing progressively towards faster methods for IM. Section 4 exhibits and interprets the experimental results for IE and IM. Finally, Section 5 summarizes the contribution and presents future steps.

## 2 RELATED WORK

The first approach to solving combinatorial optimization (CO) using neural networks was based on attention-based NNs for discrete structures, POINTERNETS (Vinyals et al., 2015), followed by an architecture that combines POINTERNETS with an actor-critic training to find the best route for TSP (Bello et al., 2016). The first architecture that utilized graph-based learning was S2N-DQN (Dai et al., 2017), using STRUCT2VEC to encode the states of the nodes and the graph, and training a Q-learning model that chooses the right node to add in a solution given the current state.

Based on S2V-DQN, a DQN (Mnih et al., 2015) for the network dismantling problem was recently proposed (Fan et al., 2020) (Li et al., 2019). The model, named FINDER, uses a deep Q-learning architecture where the representations are derived by three GRAPHSAGE layers. The reward is based on size of the giant connected component size, i.e., every new node (seed) chosen, aims to dismantle the network as much as possible. Some of the main advantages of FINDER is that it is trained on small synthetic data, which are easy to make, and can extrapolate to relatively large graphs. On the other hand, one of the core disadvantages is that it can not work with directed graphs and weighted edges. Another recent supervised deep learning approach on IM, GCOMB (Manchanda et al., 2020), utilizes a probabilistic greedy to produce scores on graphs and trains a GNN to predict them. A Q-network receives the scores along with an approximate calculation of the node's neighborhood correlation with the seed set, to predict the next seed. This approach, though scalable and comparable to SOTA in accuracy, has to be trained on a large random subset of the graph (30% of it) and tested on the rest. This makes the model graph-specific, i.e., it has to be retrained to perform well on a new graph. This imposes a serious overhead, considering the time required for training, subsampling and labeling these samples using the probabilistic greedy method with traditional IE. As shown in (Manchanda et al., 2020) Appendix G, it takes at least hundreds of minutes and is thus out of our scope. Finally, recent works on learning approximations to submodular policies **?** require a large

number of ground truth evaluation to create the training trajectories, which rendering the training too time consuming. Moreover, such methods require a novel neural network encoding to capture the state of IM, which has not been developed yet.

In this paper, we propose an approach that combines the advantages of the aforementioned methods, in that it is only trained on small simulated data once and generalizes to larger graphs, and it addresses the problem of IM in weighted directed networks. Furthermore, the approach can be broken down to a GNN for influence estimation and three IM methods. The former can act alone as influence predictor and be competitive with relevant methods, such as DMP (Lokhov & Saad, 2019) for graphs up to one scale larger than the train set. GLIE is used to propose CELF-GLIE, CELF Leskovec et al. (2007) with GLIE as influence estimator, GRIM, a Q-network that learns how to choose seeds using the GLIE's estimations and hidden representations, and PUN, an adaptive IM method that optimizes greedily a submodular influence spread using GLIE's representations.

We note here that the majority of the relevant literature on DL for CO address small graphs (Vinyals et al., 2015; Dai et al., 2017; Kool et al., 2018; Prates et al., 2019) which makes them not applicable to our task. More scalable, unsupervised methods (Karalias & Loukas, 2020) are tailored to specific problems and is non-trivial to adjust them to our problem, with the exception of (Li et al., 2018) which was found significantly worse than the SOTA algorithm we compare with in Manchanda et al. (2020).

## 3 METHODOLOGY

### 3.1 GRAPH LEARNING FOR INFLUENCE ESTIMATION (GLIE)

In this section, we introduce GLIE (**G**NN **L**earning **I**nfluence **E**stimation), which aims to learn how to estimate the influence of a seed set $S$ over a graph $G = (V, E)$. Let $\mathbf{A} \in \mathbb{R}^{n \times n}$ be the adjacency matrix and $\mathbf{X} \in \mathbb{R}^{n \times d}$ be the features of nodes, representing which nodes belong to the seed set by 1 and 0 otherwise:

$$\mathbf{X}_u = \left\{ \begin{array}{ll} \{1\}^d, & u \in S \\ \{0\}^d, & u \notin S \end{array} \right. . \tag{1}$$

For the derivation of the corollary, we set $d = 1$. More dimensions will become meaningful when we parameterize the problem. If we normalize $\mathbf{A}$ by each row, we form a row-stochastic transition matrix, as:

$$\mathbf{A}_{uv} = p_{vu} = \left\{ \begin{array}{ll} \frac{1}{\deg(u)}, & v \in \mathcal{N}(u) \\ 0, & v \notin \mathcal{N}(u) \end{array} \right. , \tag{2}$$

where $\deg(u)$ is the in-degree of node $u$ and $\mathcal{N}(u)$ is the set of neighbors of $u$. Based on the weighted cascade (Kempe et al., 2003), each row $u$ stores the probability of node $u$ being influenced by each of the other nodes that are connected to it by a directed link $v \to u$. Note that, in case of directed influence graphs, $\mathbf{A}$ should correspond to the *transpose* of the adjacency matrix. The influence probability $p(u|S)$ resembles the probability of a node $u$ getting influenced if its neighbors belong in the seed set, i.e., during the first step of the diffusion. We can use message passing to compute a well-known upper bound $\hat{p}(u|S)$ of $p(u|S)$ for node $u$:

$$\hat{p}(u|S) = \mathbf{A}_u \cdot \mathbf{X} = \sum_{v \in \mathcal{N}(u) \cap S} \frac{1}{\deg(u)} = \sum_{v \in \mathcal{N}(u) \cap S} p_{vu} \geq 1 - \prod_{v \in \mathcal{N}(u) \cap S} (1 - p_{vu}) = p(u|S), \tag{3}$$

where the second equality stems from the definition of the weighted cascade and the inequality from the proof in Zhou et al. (2015), App. A. As the diffusion covers more than one-hop, the derivation requires repeating the multiplication to approximate the total influence spread. To be specific, computing the influence probability of nodes that are not adjacent to the seed set requires estimating recursively the probability of their neighbors being influenced by the seeds. If we let $\mathbf{H}_1 = \mathbf{A} \cdot \mathbf{X}$, and we assume the new seed set $S^t$ to be the nodes influenced in the step $t - 1$, their probabilities are stored in $\mathbf{H}_t$, much like a diffusion in discrete time. We can then recompute the new influence probabilities with $\mathbf{H}_{t+1} = \mathbf{A} \cdot \mathbf{H}_t$,

**Corollary 3.0.1.** *The repeated product $\mathbf{H}_{t+1} = \mathbf{A} \cdot \mathbf{H}_t$ computes an upper bound to the real influence probabilities of each infected node at step $t + 1$.*

The proof can be found in Appendix A.1. In reality, due to the existence of cycles, two problems arise with this computation. Firstly, if the process is repeated the influence of the original seeds may increase again, which comes in contrast with the independent cascade model. This can be controlled by minimizing the repetitions, e.g., four repetitions cause the original seeds to be able to reinfect other nodes in a network with triangles. To this end, we leverage up to three neural network layers. Another problem due to cycles pertains to the probability of neighbors influencing each other. In this case, the product of the complementary probabilities in Eq. (3) does not factorize for the non-independent neighbors. This effect was analyzed extensively in Lokhov & Saad (2019), App. B, and proved that the influence probability computed by $p(u|S)$ is itself an upper bound on the real influence probability for graphs with cycles. Intuitively, the product that represents non-independent probabilities is larger than the product of independent ones. This renders the real influence probability, which is complementary to the product, smaller than what we compute.

We can thus contend that the estimation $\hat{p}(u|S)$ provides an upper bound on the real influence probability—and we can use it to compute an upper bound to the real influence spread of a given seed set i.e., the total number of nodes influenced by the diffusion. Since message passing can compute inherently an approximation of influence estimation, we can parameterize it to learn a function that tightens this approximation based on supervision. In our neural network architecture, each layer consists of a GNN with a batchnorm and dropout omitted here, and starting from $\mathbf{H}_0 = \mathbf{X} \in \mathbb{R}^{n \times d}$ we have:

$$\mathbf{H}_{t+1} = \mathrm{ReLU}([\mathbf{H}_t, \mathbf{A}\mathbf{H}_t]\mathbf{W}_0). \tag{4}$$

The readout function that summarizes the graph representation based on all nodes' representations is a summation over all the final representations with skip connections:

$$\mathbf{H}_S^G = \sum_{v \in V} [\mathbf{H}_0^v, \mathbf{H}_1^v, \dots, \mathbf{H}_t^v]. \tag{5}$$

This representation captures the probability of all nodes being active throughout each layer. The output that represents the predicted influence spread is derived by:

$$\hat{\sigma}(S) = \mathrm{ReLU}(\mathbf{H}_S^G \mathbf{W}_o). \tag{6}$$

Note that, the derived representations of each layer $H_t^i$, untrained, are the upper bound of the influence probability of seed set's the t-hop neighbors. The parameters of the intermediate layers $W_t$ are trained such that the upper bound is reduced and the final layer $W_o$ can combine the probabilities to derive a cumulative estimate for the total number of influenced nodes. We empirically verify this by examining the layer activations which can be seen in Fig. 1 and the heatmaps indicate a difference between columns (nodes) expected to be influenced, meaning we could potentially predict not only the number but also who will be influenced. However, since $\hat{\sigma}$ is derived by multiple layers, the relationships and thresholds to determine the exact influenced set is not straight forward. Below we experiment with different such sets extracted from $H$ for the purpose of IM.

## 3.2 Cost Effective Lazy Forward with Glie (Celf-Glie)

The original Cost Effective Lazy Forward (Leskovec et al., 2007) is an acceleration to the original greedy algorithm that is based on the constraint that a seed's spread will never get bigger in subsequent steps. The influence spread is computed for every node in the first iteration and kept in a sorted list. In each step, the marginal gain is computed for the node with the best influence spread in the previous round. If it is better than the previously second node, it is chosen as the next seed because it is necessarily bigger than the rest. This property stems from submodularity, i.e., the marginal gain can never increase with the size of the seed set. If the first node's current gain is smaller than the second previous gain, the list is resorted and the process is repeated until the best node is found. The worst-case complexity is similar to greedy but in practice it can be hundreds of times faster, while retaining greedy's original guarantee.

In our case, we propose a straight forward adaptation where we substitute the original CELF IE based on MC IC with the output of GLIE. We redirect the reader in the Appendix B, where we show that the estimations of GLIE are monotonous and submodular in practice, and hence $\hat{\sigma}$ is suitable for optimizing with CELF. Since we do not prove the submodularity of $\hat{\sigma}$, we can not contend that the theoretical guarantee is retained. CELF-GLIE has two main computational bottlenecks. First,

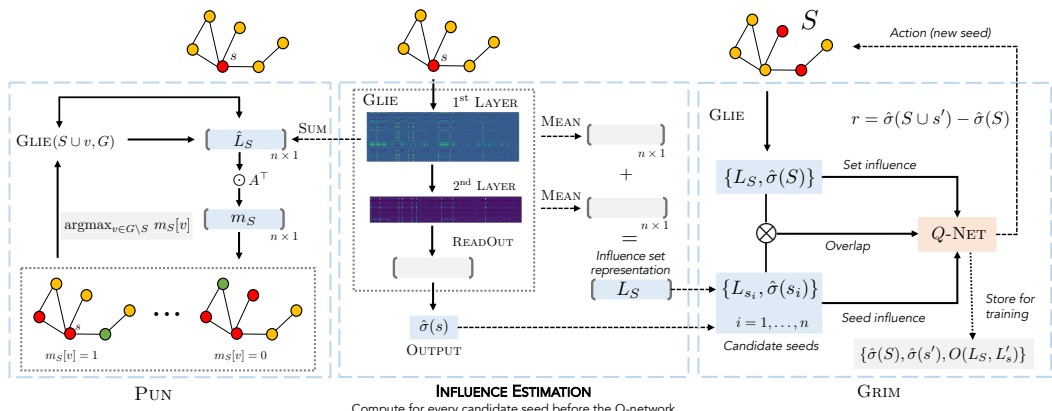

Figure 1: A visual depiction of the pipeline for GRIM and PUN. The layers of GLIE are depicted by a heatmap of an actual seed during inference time, showing how the values vary through different nodes (columns).

although it alleviates the need to test every node in every step, in practice it still requires IE for more than one nodes in each step. Second, it requires computing the initial IE for every node in the first step. We will try to alleviate both with the two subsequent methods.

### 3.3 GRAPH REINFORCEMENT LEARNING FOR INFLUENCE MAXIMIZATION (GRIM)

We develop a method that computes only one IE in every step, along with the initial IE for all nodes. We first utilize the activations mentioned above to define the influence set representation $L_S \in \{0,1\}^n$, which can be computed by adding the activations of each layer $\mathbf{H}_t$, summing along the axis of the hidden layer size, and thresholding to get a binary vector:

$$L_S = \mathbb{1}\left\{ \sum_{t=0}^{T} \frac{\sum_{i=0}^{d_t} \mathbf{H}_t^i}{d_t} \geq 0 \right\}, \tag{7}$$

where $T$ is the number of layers, and $\mathbf{H}_t^i \in \mathbb{R}^{n \times 1}$ is a column from $\mathbf{H}_t$. This vector contains a label for each node whose sign indicates if it is predicted to be influenced. We compute the average representation because $d_t$ varies throughout layers, and since we add all layer's outputs, we need an equal contribution from each layer's dimension to the final output. We utilize this to compute the difference between the influence of the current seed set and the initial influence of each other node.

We aim to build a method that learns how to pick seeds sequentially. The model needs to receive information from GLIE regarding the state (graph and seed set), and decide on the next action (seed). Note that, GLIE can not provide a direct estimate of a new candidate's $s$ marginal gain without rerunning $\text{GLIE}(S \cup s, G)$, which is what we try to avoid. To this end, we utilize a double Q-network (Van Hasselt et al., 2016) and the model is depicted in Fig. 1 (middle and right part). During the first step, GLIE provides an IE for all candidate seeds, and the node with the highest is added to the seed set, similar to GLIE-CELF. We also keep a list of each node's initial influence set $L_s$. Subsequently, the Q-network produces a Q-value for each node $s$ using as input the estimated influence of the current seed set $\hat{\sigma}(S)$, the initial influence of the node $\hat{\sigma}(s)$, and the interaction between them. The interaction is defined as the difference between their corresponding influence sets $O(S, s) = \sum_{i=0}^{n} \mathbb{1}\{L_s^i - L_S^i \geq 0\}$, as predicted by GLIE. The latter aims to measure how different is the candidate node from the seed set, in order to quantify the potential gain of adding it. The Q-network is called **G**raph **R**einforcement for **I**nfluence **M**aximization (GRIM), and its architecture is composed by two layers:

$$Q(u, S, G) = \text{ReLU}(\text{ReLU}([\hat{\sigma}_S, \hat{\sigma}_s, O(S, s)]\mathbf{W}_k)\mathbf{W}_q), \tag{8}$$

where $\mathbf{W}_q \in \mathbb{R}^{hd \times 1}$, and $hd$ is the hidden layer size. We utilize a greedy policy to choose the next seed, similar to Dai et al. (2017): $\pi(u|S) = \arg\max_{u \in S} Q(u, S, G)$. Given the chosen action $u$, the reward is computed based on the marginal gain, i.e., the estimated influence of the new seed set minus the influence of the seed set before the action, as computed by GLIE in:

$$r = \hat{\sigma}(S \cup u) - \hat{\sigma}(S). \tag{9}$$

During training, we use epsilon-greedy to simulate an IM "game" and balance between exploration and exploitation. We store as a train tuple the current state-action embedding $\{\hat{\sigma}(S), \hat{\sigma}(s'), O(S, s')\}$, the new state embedding along with all next possible actions $\{\hat{\sigma}(S \cup s), \hat{\sigma}(s), O(S, s)\}, s \in V$ and the reward $r$. Throughout the IM, we randomly sample from the memory and train the parameters of GRIM. GLIE is frozen, because apart from providing graph and node embeddings, it is only used for the computation of the reward—thus, it is not updated. The strategy to pretrain a graph encoding layer on a supervised task and use it as part of the Q-network has proven beneficial in similar works (Mirhoseini et al., 2021). In our case though we found out that without the overlap $O$ and with simpler features such as the degree, the model performs rather poorly, which means we can not alleviate the burden of computing IE for every node in the first step.

### 3.4 POTENTIALLY UNINFLUENCED NEIGHBORS (PUN)

Computing the influence spread of every node in the first step is too computationally demanding. We thus seek a method that can surpass this hinder and provide adequate performance. We first need to redefine a simpler influence set representation then in 7. Let $\hat{L}_S, L'_S \in \{0, 1\}^n$ be the binary vectors with 1s in nodes predicted to be uninfluenced and nodes predicted to be influenced respectively:

$$\hat{L}_S = \mathbb{1}\left\{\sum_{i=0}^{d_1} \mathbf{H}_1^i \le 0\right\} \quad L'_S = \mathbb{1}\left\{\sum_{i=0}^{d_1} \mathbf{H}_1^i \ge 0\right\} \tag{10}$$

$L'_S$ is simpler than $L_S$ defined in Eq. (7) and provides a more rough estimate, but it allows for a simpler influence spread which we can optimize greedily.

$$\sigma^m(S) = |L'_S| \tag{11}$$

We can use $\hat{L}_S$ and message passing to predict the amount of a node's neighborhood that remains uninfluenced, i.e., the **P**otentially **U**ninfluenced **N**eighbors (PUN), weighted by the respective probability of influence For a node $u$,

$$m_S[u] = \sum_{v \in N(u)} A_{u,v}\hat{L}_v = A_u^\top \cdot \hat{L}_S \in \mathbb{R}^{n \times 1}. \tag{12}$$

For efficiency, we can compute $m_S = A^T\hat{L}$ which can be considered an approximation to all nodes marginal gain on their immediate neighbors. We can thus optimize this using $argmax(m_S)$, as shown in Fig. (1). In order to establish that $\sigma^m$ can be optimized greedily with a theoretical guarantee of $(1 - \frac{1}{e})$OPT, we prove its monotonocity and submodularity in Appendix A.2.

**Theorem 1.** *The influence spread $\sigma^m$ is submodular and monotone.*

PUN can be seen in the left part of Fig. 1. We start by setting the first seed as the node with the highest degree, which can be considered a safe assumption as in practice it is always part of seed sets. We use GLIE($S, G$) to retrieve $\hat{L}_S$, which we use to find the next node based on $\arg\max_{v \in G \setminus S} m_S[v]$ and the new $\hat{L}_{S \cup \{v\}}$. One disadvantage of PUN is that $\sigma^m$ is an underestimation of the predicted influence, as can be seen in Fig. 2. Contrasted with the upper bound, DMP, $\sigma^m$ is not as accurate as $\hat{\sigma}$, but allows us to compute efficiently a submodular proxy for the marginal gain. This underestimation means that a part of the network considered uninfluenced in $\hat{L}_S$ is measured as potential gain for their neighbors, hence the ranking based on $m_S$ can be effected negatively. As we observed in Fig. 2, the divergence of $\sigma^m$ increases with the size of the seed set.

For this purpose, we will use adaptive full-feedback selection (AFF), where after selecting a new seed node, we remove it from the network along with nodes predicted to be influenced. It has been proved in the seminal work of Golovin & Krause (2011) that an AFF greedy algorithm for a submodular and monotonic function is guaranteed to have a competitive performance with the optimal policy. In our case, we will use an AFF update every $k$ seeds, as it adds a small computational overhead if we do it in every step. The benefit to PUN is twofold. Firstly, as we remove the influenced node and truncate the seed set, GLIE produces a more valid estimate because it performs better when the graph and seed set are smaller. Secondly, as the neighborhood size decreases, the effect of missed influenced nodes is diminished in $m_S$. These can be observed in the "Adaptive" plots of Fig. 2 (b), (d) and in other related figures in Appendix B, where we employ an AFF every 10 seeds and we contrast the aforementioned gap between the upper bound DMP and $\sigma^m(S)$.

## 4 EXPERIMENTS

All the experiments are performed in a PC with an NVIDIA GPU TITAN V (12GB RAM), 256GB RAM and an Intel(R) Xeon(R) W-2145 CPU @ 3.70GHz. The python implementation can be found in the supplementary material.

### 4.1 INFLUENCE ESTIMATION

For training in the influence estimation task, we create a set of labeled samples, each consisting of the seed set $S$ and the corresponding influence spread $\sigma(S)$. We create 100 Barabasi-Albert Barabási & Albert (1999) and Holme-Kim Holme & Kim (2002) undirected graphs ranging from 100 to 200 nodes and 30 from 300 to 500 nodes. 60% are used for training, 20% for validation and 20% for testing. We have used these network models because the degree distribution resembles the one of real world networks. The influence probabilities are assigned based on the weighted cascade model, i.e., a node $u$ has equal probability $1/\deg(u)$ to be influenced by each of her $\mathcal{N}(u)$ nodes. This model requires a directed graph, hence we turn all undirected graphs to directed ones by appending reverse edges. Though estimating influence probabilities is a problem on its own (Panagopoulos et al., 2020; Du et al., 2014), in the absence of extra data, the weighted cascade is considered more realistic than pure random assignments (Kempe et al., 2003). To label the samples, we run the CELF algorithm using $1,000$ Monte Carlo (MC) ICs for influence estimation, for up to 5 seeds. The optimum seed set for size 1 to 5 is stored, along with 30 random negative samples for each seed set size. Each sample is accompanied with its ground truth influence spread computed with MC ICs. This amounts to a total of $20,150$ training samples. More details about the training parameters are presented in Appendix B.1.

We evaluate the models in three different types of graphs. The first is the test set of the dataset mentioned above. The second is a set of 10 power-law large graphs $(1,000-2,000$ nodes) to evaluate the capability of the model to generalize in networks that are larger by one factor. The third is a set of three real-world graphs, namely the *Crime* (CR), *HI-II-14* (HI), and *GR collaborations* (GR). More information about the datasets is given in Table 1. The

|  | Graph | # Nodes | # Edges |
|---|---|---|---|
| Small Sim | Test/Train | $100-500$ | $950-4,810$ |
| | Large | $1,000-2,000$ | $11,066-19,076$ |
| Small | Crime (CR) | $829$ | $2,946$ |
| | HI-II-14 (HI) | $4,165$ | $26,172$ |
| | GR Colab (GR) | $5,242$ | $28,980$ |
| Large | Enron (EN) | $33,697$ | $361,622$ |
| | Facebook (FB) | $63,393$ | $1,633,660$ |
| | Youtube (YT) | $1,134,891$ | $5,975,246$ |

Table 1: Graph datasets.

real graphs are evaluated for varying seed set sizes, from 2 to 10, to test our model's capacity to extrapolate to larger seed set sizes. Due to the size of the latter two graphs (HI and GR), we take for each seed set size the top nodes based on the degree as the optimum seed set along with a 30 random seed sets for the large simulated graphs and 3 for the real graphs, to validate the accuracy of the model in non-significant sets of nodes.

We have compared the accuracy of influence estimation with DMP (Lokhov & Saad, 2019). We could not utilize the influence estimation of UBLF (Zhou et al., 2015) because its central condition

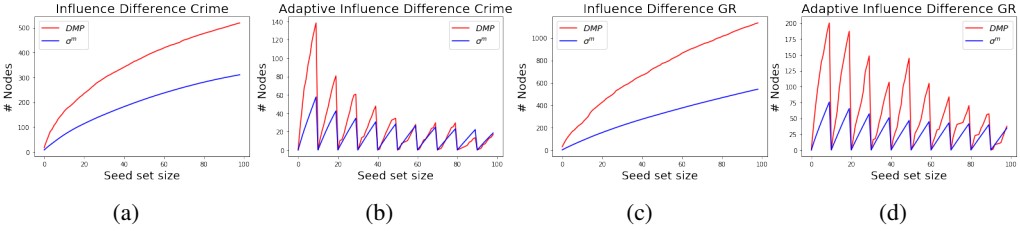

(a)  (b)  (c)  (d)

Figure 2: Difference between DMP influence estimate and $\sigma^m$ in standard IM and adaptive IM with full feedback every 10 seeds, in two datasets (Crime and GR).

| Graph | DMP | | GLIE | |
|---|---|---|---|---|
| (seeds) | **MAE** | **Time** | **MAE** | **Time** |
| Test $(1-5)$ | 0.076 | 0.05 | 0.046 | 0.0042 |
| Large $(1-5)$ | 0.086 | 0.44 | 0.102 | 0.0034 |
| CR $(1-10)$ | 0.009 | 0.11 | 0.044 | 0.0029 |
| HI $(1-10)$ | 0.041 | 2.84 | 0.056 | 0.0034 |
| GR $(1-10)$ | 0.122 | 4.32 | 0.084 | 0.0042 |

Table 2: Average mean absolute error (MAE) devided by the average influence and time (in seconds) throughout all seed set sizes and samples, along with the real average influence spread.

| Graph | Seed | DMP-CELF | | GLIE-CELF | |
|---|---|---|---|---|---|
| (seeds) | **Overlap** | **Influence** | **Time** | **Influence** | **Time** |
| CR(20) | 14 | 221 | 83 | 229 | 1.0 |
| HI(20) | 13 | $1,235$ | $8,362$ | $1,281$ | 5.49 |
| GR(20) | 12 | 295 | $16,533$ | 393 | 7.01 |

Table 3: Influence maximization for **20** seeds with CELF, using the proposed (GLIE) substitute for influence estimation and evaluating with $10,000$ MC independent cascades (IC).

is violated by the weighted cascade model and the computed influence is exaggerated to the point it surpasses the nodes of the network. The average error throughout all datasets and the average influence can be seen in Table 2, along with the average time. We evaluate the retrieved seed set using the independent cascade, and the results are shown in Table 3. We should underline here that this task would require more then 3 hours for the *Crime* dataset and days for *GR* using the traditional approach with $1,000$ MC IC. As we can see in Table 3, GLIE-CELF allows for a significant acceleration in computational time, while the retrieved seeds are more effective. Moreover, in CELF, the majority of time is consumed in the initial computation of the influence spread, i.e., the overhead to compute 100 instead of the 20 seeds shown in Table 3, amounts to $0.11$, $0.22$ and $0.19$ seconds for the three datasets respectively.

## 4.2 INFLUENCE MAXIMIZATION

The training parameters for GRIM are analyzed in Appendix B (B.1). For comparison, we use a state-of-the-art IM method, IMM (Tang et al., 2015) which capitalizes on reverse reachable sets (Borgs et al., 2014) to estimate influence. Specifically, it produces a series of such influence sketches and uses them to approximate the influence spread without any simulation when building the seed set. This results in remarkable acceleration while retaining a theoretical guarantee with high probability. Note that, IMM is considered state-of-the-art and has similar influence spreads with Tang et al. (2014), while surpassing various heuristics (Jung et al., 2012). We set $e = 0.5$ as proposed by the authors. We also compare with FINDER, which is analyzed in Section 2 , and with the most well known heuristic methods for the Independent Cascade PMIA (Wang et al., 2012), DEGREEDISCOUNT (Chen et al., 2009) and K-CORES (Malliaros et al., 2020) .

| Graph | GLIE-CELF | GRIM | PUN | K-CORE | PMIA | DEGDISC | IMM | FINDER |
|---|---|---|---|---|---|---|---|---|
| CR | **522** | 509 | 521 | 455 | 520 | 512 | 516 | 502 |
| GR | **1,102** | 997 | $1,076$ | 421 | $1,013$ | 919 | $1,085$ | 897 |
| HI | $2,307$ | $1,302$ | **2,308** | $2,024$ | $2,291$ | $2,229$ | $2,290$ | $2,274$ |
| EN | **14,920** | $14,022$ | $14,912$ | $10,918$ | $14,855$ | $13,808$ | $14,848$ | $12,596$ |
| FB | **8,710** | $7,418$ | $8,409$ | $4,174$ | $5,613$ | $8,247$ | $8,625$ | $5,746$ |
| YT | $189,515$ | $187,808$ | $187,808$ | $89,546$ | $189,746$ | **194,834** | $194,521$ | $34,941$ |

Table 4: Influence spread computed by 10,000 MC ICs for 100 seeds.

The results for the influence spread of 100 and 200 seeds as computed by simulations of MC ICs can be seen in Table (4) and (5), while the time results are shown in Table 6. The top result is in bold and the second best is underlined. The influence spread for smaller seed sets and the time for heuristics are attached in Appendix C, along with comparisons of PUN without the use of GPU. One can see

| Graph | GLIE-CELF | GRIM | PUN | K-CORE | PMIA | DEGDISC | IMM | FINDER |
|---|---|---|---|---|---|---|---|---|
| CR | **661** | 650 | 657 | 647 | 656 | 644 | 650 | 642 |
| GR | 1,617 | 1,502 | **1,626** | 701 | 1,566 | 1415 | 1,617 | 1,286 |
| HI | 2,685 | 2,631 | **2,688** | 2,540 | 2,685 | 2,614 | 2,668 | 2,625 |
| EN | 17,601 | 16,642 | **17,614** | 13,015 | 17,534 | 16,500 | 17,497 | 17,244 |
| FB | 10,981 | 9,406 | 10,626 | 6,434 | 7,688 | 10,309 | **11,007** | 10,801 |
| YT | **246,439** | 241,000 | 244,579 | 110,409 | 242,057 | 236,726 | 247,178 | 50,435 |

Table 5: Influence spread computed by 10,000 MC ICs for 200 seeds.

| | 100 **seeds** | | | | | 200 **seeds** | | | | |
|---|---|---|---|---|---|---|---|---|---|---|
| Graph | GLIE-CELF | GRIM | PUN | IMM | FINDER | GLIE-CELF | GRIM | PUN | IMM | FINDER |
| CR | 1.25 | 0.91 | 0.15 | **0.13** | 0.41 | 2.00 | 2.03 | 0.25 | **0.19** | 0.41 |
| GR | 3.41 | 0.69 | **0.17** | 0.57 | 2.36 | 4.55 | 1.79 | **0.26** | 0.95 | 2.36 |
| HI | 1.20 | 2.59 | **0.17** | 0.56 | 1.01 | 2.19 | 0.60 | **0.27** | 1.29 | 1.01 |
| EN | 5.89 | 4.85 | **0.52** | 4.78 | 9.30 | 15.49 | 5.49 | **0.97** | 10.47 | 9.30 |
| FB | 120.6 | 100.00 | **1.42** | 69.90 | 56.8 | 287.7 | 123.95 | **3.1** | 171.25 | 56.80 |
| YT | 119.00 | 48.00 | **13.20** | 55.40 | 191.00 | 151.33 | 100.00 | **28.92** | 82.13 | 191.00 |

Table 6: Computational time in seconds.

that GLIE-CELF exhibits overall superior influence quality compared to the rest of the methods, but is quite slower. GRIM is slightly faster than GLIE-CELF but is the second slowest method. This quantifies the substantial overhead caused by computing the influence spread of all candidate seeds in the first step. Their time difference amounts to how many more IEs GLIE-CELF performs in every step compared to GRIM, which performs only one. This is more obvious with PUN, which requires only one IE in every step and no initial computation. It is from 3 to 60 times faster than IMM while its computational overhead moving from smaller to larger graphs is less than linear to the number of nodes. In terms of influence quality, PUN is first or second in the majority of the datasets in both seed set sizes. We can thus contend that it provides the best accuracy-efficiency tradeoff from the examined methods. IMM is the third slowest method, but it is very accurate, specially for smaller seed set sizes. FINDER exhibits the least accurate performance, which is understandable given that it solves a relevant problem and not exactly IM for IC. The computational time presented is the time required to solve the node percolation, in which case it may retrieve a bigger seed set than 100 nodes. Thus, we can hypothesize it is quite faster for a limited seed set, but the quality of the retrieved seeds is the least accurate among all methods.

## 5 CONCLUSION

We have proposed GLIE, a GNN-based solution for the problem of influence estimation. We showcase its accuracy in that task and utilized it to address the problem of influence maximization. We developed three methods based on the representations and the predictions of GLIE. GLIE-CELF, an adaptation of a classical algorithm that surpasses SOTA but with significant computational overhead. GRIM, a Q-learning model that learns to retrieve seeds sequentially using GLIE's predictions and representations. And PUN, a submodular function that acts as proxy for the marginal gain and can be optimized adaptively, striking a balance between efficiency and accuracy.

A typical IM algorithm needs a significant contribution in order to take into account the topic of the information shared or the user's characteristics (Chen et al., 2016) i.e., conditional diffusion. An important practical advantage of a neural network approach is the easy incorporation of such complementary data by adding the corresponding embeddings in the input, as has been done in similar settings (Tian et al., 2020). We thus deem an experiment with contextual information a natural next step, given a proper dataset. Our approach can also be utilized to address the minimum vertex cover in large graphs, as it is a problem related to influence maximization and there exists models that work well in both (Manchanda et al., 2020). Finally, we also plan to examine the potential of training online the reinforcement learning, i.e., receiving real feedback from each step of the diffusion that could update both, the Q-NET and GLIE. This would allow the model to adjust its decisions based on the partial feedback received during the diffusion.

**Ethics and Reproducibility** Our influence maximization methods can potentially be used for large scale manipulation in social media. For example, targeting the right susceptible users with specific political advertisements might maximize the effect of the campaign on the general intent to vote. On the bright side, there has been extensive research on battling such effects (Tu et al., 2020), by producing campaigns that are equally effective and hence balancing each other (Bharathi et al., 2007). More importantly, GLIE can be used as part of the mitigation strategy (Tu et al., 2020), as a black box that substitutes the method's RR-set based influence estimation, to actually increase the balance of political exposure in the given social network for the running campaigns. Moreover, our methods can inherently be utilized in the context of limiting fake news spreading similar to other influence maximization algorithms (Budak et al., 2011).

Regarding the reproducibility of our experiments, we have attached all codes in the supplementary files along with detailed instructions on how to reproduce the results. We also point to the codes we utilized to run the benchmarks. All data can be downloaded through an anonymous repository.

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

SUPPLEMENTARY MATERIAL:
LEARNING GRAPH REPRESENTATIONS FOR INFLUENCE MAXIMIZATION

Appendix A gathers the proofs of theorems and corollaries, appendix B includes the training details and plots in support of the arguments in the main text, and finally, appendix C shows extra results.

APPENDIX A

We recall the first corollary from Sec. 3 and detail the proof below.

**Corollary 3.0.2.** *The repeated product* $\mathbf{H}_{t+1} = \mathbf{A} \cdot \mathbf{H}_t$ *computes an upper bound to the real influence probabilities of each infected node at step* $t + 1$.

*Proof.* We have:

$$\hat{p}^t(u|S^t) = \mathbf{A}_u \cdot \mathbf{H}_t \tag{13}$$

$$= \sum_{v \in \mathcal{N}(u) \cap S^t} \hat{p}_v p_{vu} \tag{14}$$

$$\geq \sum_{v \in \mathcal{N}(u) \cap S^t} p_v p_{vu} \tag{15}$$

$$\geq 1 - \prod_{v \in \mathcal{N}(u) \cap S^t} (1 - p_v p_{vu}) \tag{16}$$

$$= p^t(u|S^t) \tag{17}$$

$$\tag{18}$$

- (15) stems from Eq. 3 in the manuscript:

$$\hat{p}(u|S) = \mathbf{A}_u \cdot \mathbf{X} = \sum_{v \in \mathcal{N}(u) \cap S} \frac{1}{\deg(u)} = \sum_{v \in \mathcal{N}(u) \cap S} p_{vu} \geq 1 - \prod_{v \in \mathcal{N}(u) \cap S} (1 - p_{vu}) = p(u|S). \tag{19}$$

- (16) can be proved by induction similar to Zhou et al. (2015). For every $p_v \leq 1$, the base case $\sum_{v \in \mathcal{X}} p_v p_{vu} \geq 1 - \prod_{v \in \mathcal{X}} (1 - p_v p_{vu})$ is obvious for $|\mathcal{X}| = 1$. For $|\mathcal{X}| > 1$, we have:

$$1 - \prod_{v \in \mathcal{X}} (1 - p_v p_{vu}) = 1 - (1 - p_x p_{xu}) \prod_{v \in \mathcal{X} \setminus x} (1 - p_v p_{vu})$$

$$= 1 - \prod_{v \in \mathcal{X} \setminus x} (1 - p_v p_{vu}) + p_x p_{xu} \prod_{v \in \mathcal{X} \setminus x} (1 - p_v p_{vu})$$

$$\leq \sum_{v \in \mathcal{X} \setminus x} p_v p_{vu} + p_x p_{xu} \prod_{v \in \mathcal{X} \setminus x} (1 - p_v p_{vu})$$

$$\leq \sum_{v \in \mathcal{X} \setminus x} p_v p_{vu} + p_x p_{xu}$$

$$= \sum_{v \in \mathcal{X}} p_v p_{vu}. \tag{20}$$

- (17) we have $p(u|v) = p_v p_{vu}$ per definition of the independent cascade, and consequently $p(u|S) = 1 - \prod_{v \in \mathcal{N}(u) \cap S} (1 - p_v p_{vu})$, where $p_v = 1$ for $v \in S^1$, which are the initial seed set that are activated deterministically. We can thus contend that Eq. equation 18 stands, and the computed probabilities are an upper bound of the real influence probabilities. Hence the influence spread, which is computed as $\hat{\sigma}(S) = \sum_{(u,v) \in E} p_{uv}$ is also an upper bound to the real $\sigma(S)$.

$\square$

Now we address the Theorem from Sec. 3.4 and detail the proof below.

**Theorem 1.** *The influence spread $\sigma^m$ is submodular and monotone.*

For the purposes of the proof, $X_i \in \{0,1\}^{n \times d}$ is the input and $H_i \in \mathbb{R}^{n \times hd}$ is the output of the first neural layer for the input seed set $S_i$, and $P \in \{1\}^{hd \times 1}$. Moreover we define the support function $supp(v) = \{i \in [1, n], v_i \neq 0\}$ James et al. (2013) as the set of indices of non zero rows in a matrix such as $X_i$. Finally let $R$ represent ReLU and $b_{tr}, st_{tr}$ the mean and standard deviation computed by the batchnorm. Each step is justified further below.

*Proof.* Monotonocity, $\forall i < j, S_i \subset S_j$:

$$supp(X_j) \supset supp(X_i) \tag{21}$$

$$supp(X_j W) \supseteq supp(X_i W) \tag{22}$$

$$supp(A X_j W) \supseteq supp(A X_i W) \tag{23}$$

$$supp(R(A X_j W)) \supseteq supp(R(A X_i W)) \tag{24}$$

$$supp\left(\frac{R(A X_j W) - b_{tr}}{st_{tr}}\right) \supseteq supp\left(\frac{(R(A X_i W) - b_{tr})}{st_{tr}}\right) \tag{25}$$

$$supp(H_j) \supseteq supp(H(S_i)) \tag{26}$$

$$supp(H_j P) \supseteq supp(H_i P) \tag{27}$$

$$|\mathbb{1}_{>0}\{H_j P\}| \geq |\mathbb{1}_{>0}\{H_i P\}| \tag{28}$$

$$|L_j'| \geq |L_i'| \tag{29}$$

$$\sigma^m(S_j) \geq \sigma^m(S_i) \tag{30}$$

$$\tag{31}$$

1. (21) stems by the definition of $X$ in Eq. (1).

2. (22) $X_j$ is a convex hull that contains $X_i$ (Boyd et al., 2004). We multiply both sides by a real matrix $W \in \mathbb{R}^{d \times hd}$ which can equally dilate both convex hulls in terms of direction and norm. This equal transformation cannot change the sign of the difference between the elements of $X_i$ and $X_j$ and hence cannot interfere with the support of $X_j$ over $X_i$. The statement becomes more obvious for $X \in \{0,1\}^{n \times 1}$ and $W \in \mathbb{R}^{1 \times 1}$. Note that both can result in zero matrices so we use subset or equal.

3. (23) $A$ is a non-negative matrix.

4. (24) ReLU is a non negative monotonically increasing function.

5. (25) Subtract by the same number and divide by the same positive number.

6. (26) Definition in Eq. (4).

7. (27) $P$ is positive.

8. (28) By definition of the support.

9. (29) By definition of $L_S'$.

□

For the proof of submodularity we have to define $X_{iu} = X_{S_i \cup u}, u \in V$ and note by the definition of the input that $|X_{ju} - X_j| = |X_{iu} - X_i|$ for the $l_1$ norm (sum of all elements):

*Proof.* Submodularity $\forall i < j, S_i \subset S_j, ,$:

$$|X_{ju} - X_j| = |X_{iu} - X_i| \tag{32}$$
$$A|X_{ju} - X_j| = A|X_{iu} - X_i| \tag{33}$$
$$|A(X_{ju} - X_j)| = |A(X_{iu} - X_i)| \tag{34}$$
$$|AX_{ju} - AX_j| = |AX_{iu} - AX_i| \tag{35}$$
$$|AX_{ju}W - AX_jW| = |AX_{iu}W - AX_iW| \tag{36}$$
$$R(|AX_{ju}W - AX_jW|) - 2b_{tr} = R(|AX_{iu}W - AX_iW| - 2b_{tr}) \tag{37}$$
$$|R(AX_{ju}W) - R(AX_jW) - 2b_{tr}| = |R(AX_{iu}W) - R(AX_iW) - 2b_{tr}| \tag{38}$$
$$supp(R(AX_{ju}W) - R(AX_jW) - 2b_{tr}) = supp(R(AX_{iu}W) - R(AX_iW) - 2b_{tr}) \tag{39}$$
$$supp(R(AX_{ju}W - b_{tr})) - supp(R(AX_jW) - b_{tr}) \subseteq supp(R(AX_{iu}W - b_{tr})) - supp(R(AX_iW) - b_{tr}) \tag{40}$$
$$supp(H_{ju}) - supp(H_j) \subseteq supp(H_{iu}) - supp(H_i) \tag{41}$$
$$\sigma^m(S^j \cup \{u\}) - \sigma^m(S^j) \le \sigma^m(S^i \cup \{u\}) - \sigma^m(S^i) \tag{42}$$
$$\tag{43}$$

1. Distributive property

2. (36) Similar to multiplication by A.

3. (40) The norm of the difference is distributed equally, but the right hand difference has as least the same or more positive elements because the norm of $A$ ,which is stochastic ,is bounded by $V$ hence $X_u$ can give up to the same gain to $AX_j$ and $AX_i$, the same number $b_{tr}$ is subtracted, and more elements are activated by $X_j$ then $X_i$ as shown in Eq. 29.

4. (41) We skipped dividing by $st_{tr}$ for brevity.

5. (42) Arrive with similar steps as 27 - 30.

$\square$

Regarding the approximation of the marginal gain we first show that choosing the node corresponding to the maximum $m_S$ will give the maximum $L'_{ju}$: $A'_u \hat{L}_j \ge A'_v \hat{L}_i \Rightarrow L'_{ju} \ge L'_{iv}$.

$$A'_u \hat{L}_j = \sum_{v \in N(u)} A'_{uv} \hat{L}_j[v] = \sum_{v \in N(u)} A_{uv} L'_j[u] = \sum_{v \in N(u)} A_{uv} X_{ju} \tag{44}$$

$$\tag{45}$$

This means that $m_S$ gives the node $u$ that improves the biggest number of rows in $AX_{ju}$ that are not already considered influenced. Since we know from eq. 30 that a $AX_{iu} \ge AX_{iv} \Rightarrow |L'_{iu}| \ge |L'_{iv}|$, the claim concludes. Hence choosing the best node using the marginal gain approximation is as good as the real influence spread. Now we prove the submodularity of the proposed marginal gain.

*Proof.* Submodularity for the approximation of the marginal gain, $\forall i < j, S_i \subset S_j$, starting from (28):

$$|\mathbb{1}_{>0}\{H_jP\}| \ge |\mathbb{1}_{>0}\{H_iP\}|$$
$$|\mathbb{1}_{\le 0}\{H_jP\}| \le |\mathbb{1}_{\le 0}\{H_iP\}| \tag{46}$$
$$A'_u \hat{L}_j \le A'_u \hat{L}_i \tag{47}$$
$$m_{S_j}[u] \le m_{S_i}[u] \tag{48}$$
$$(|L'_j| + m_{S_j}[u]) - |L'_j| \le (|L'_i| + m_{S_i}[u]) - |L'_i| \tag{49}$$
$$\sigma^m(S^j \cup \{u\}) - \sigma^m(S^j) \le \sigma^m(S^i \cup \{u\}) - \sigma^m(S^i) \tag{50}$$

1. (30) Complementarity between elements that are $\leq 0$ and elements $> 0$.

2. (32) Definition in Eq. (10) and multiply with non-negative row $u$ from matrix $A'$.

3. (33) Definition in Eq. ( 12).

4. (34) Adding and subtracting $|L_j|$ and $|L_i|$.

5. (1) By definition of $\sigma^m$ in Eq. (11) and the marginal gain of $u$, we arrive at submodularity in Eq. (1).

$\square$

## APPENDIX B

### B.1 TRAINING

As mentioned in the main text each training sample for GLIE corresponds to a triple of a graph $G$ a seed set $S$ and a ground truth influence spread $\sigma(S)$ that serves as a label to regress on. We use two different ways to come up with the seed sets $S$ for varying sizes of $S$ from 1 to 5. We use random seed sets in order to capture the average influence spread expected for a seed set of about that size. This creates "average samples" which would constitute the whole dataset in other problems. In IM however, the difference in $\sigma$ between an average seed set and the optimum seed set can be significant, hence training solely on the random sets would render our model unable to predict larger values that correspond to the optimum. We thus add in the samples the optimum seed set for each size, taken using **Celf** and MC ICs for influence estimation. For each size, we have the 30 random seed sets and the optimum, which is a more balanced form of supervision, as you expect the crucial majority of the seed sets to have an average $\sigma$. Regarding the training procedure, we have used a small scale grid-search using the validation set to find the optimum batch size $64$, dropout $0.4$, number of layers $2$, hidden layer size $64$, and feature dimension $50$. More importantly, we observed that it is beneficial to decrease the hidden layer size (by a factor of 2) as the depth increases, i.e., go from 32 to 16. This means that the 1-hop node representations are more useful compared to the 2-hop ones and so on—validating the aforementioned conclusion that the approximation to the influence estimation in Eq. (3), diverges more as the message passing depth increases. The training then proceeds for 100 epochs with an early stopping of 50 and learning rate of $0.01$.

GRIM is trained on a dataset that consists of 50 random BA graphs of $500 - 2,000$ nodes. It is trained by choosing 100 seeds sequentially, to maximize the reward (delay = 2 steps) for each network. Since the immediate reward corresponds to the marginal gain, the sum of these rewards at the end of the "game" corresponds to the total influence of the seed set. An episode corresponds to completing the game for all 50 graphs; we play 500 episodes, taking roughly $40$ seconds each. The exploration is set to 0.3 and declines with a factor of 0.99. The model is optimized using ADAM, as in GLIE. We store the model that has the best average influence over all train graphs in a training episode. In order to diminish the computational time of the first step in GLIE-CELF and GRIM, we focus on candidate seeds that surpass a certain degree threshold based on the distribution, a common practice in the literature (Chen et al., 2009; Manchanda et al., 2020).

## B.2 ADAPTIVE FULL FEEDBACK

We see in Fig. 3 that the pattern that is prevalent in the two datasets described in Section 3.4 is retained in other graphs.

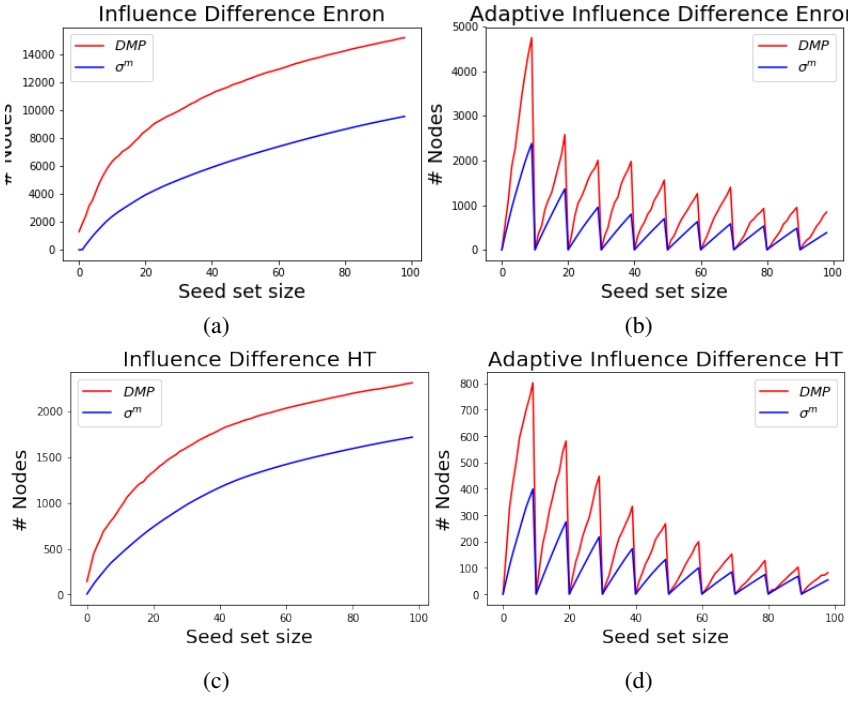

(a)               (b)

(c)               (d)

Figure 3: Difference between DMP influence estimate and $\sigma^m$ in regular and adapting with full feedback every 10 seeds.

## B.3 GLIE SUBMODULARITY AND MONOTONICITY

In this section we design an experiment to empirically prove that GLIE's output is submodular and monotonous. For each of the real datasets, we use the seed set retrieved by GLIE-CELF and a random seed set to quantify the differences between subsequent estimations. To be specific, we have a sequence $S$ that represents the seed set and a sequence $R$ that represents the random nodes, with $S_j$ being the seed set up to $j^{th}$ element and $s_j$ being the $j^{th}$ element, and similarly $r_j$ for $R$. We compute the marginal gain to check for monotonicity:

$$m_{ss} = \hat{\sigma}(S_j \cup s_{j+1}) - \hat{\sigma}(S_j) \tag{51}$$

$$m_{sr} = \hat{\sigma}(S_j \cup r_{j+1}) - \hat{\sigma}(S_j), \tag{52}$$

and for submodularity, we have , with $i = j - 1$ :

$$s_{ss} = (\hat{\sigma}(S_i \cup s_{j+1}) - \hat{\sigma}(S_i)) - (\hat{\sigma}(S_j \cup s_{j+1}) - \hat{\sigma}(S_j)) \tag{53}$$

$$s_{sr} = (\hat{\sigma}(S_i \cup r_{j+1}) - \hat{\sigma}(S_i)) - (\hat{\sigma}(S_j \cup r_{j+1}) - \hat{\sigma}(S_j)). \tag{54}$$

In Figures 4 to 7, we plot $m$ and $s$ for some of our datasets. Regarding $s$, since we require a constant node, we randomly sample one of the seeds $s_j$ and a random node $r_j$ and visualize the sequences of both $s$ with regard to adding them in every step. The values of these functions correspond to nodes, and range from tens to thousands, depending on the datasets. For monotonicity and submodularity, we verify that $m$ and $s$ are always more than zero. Moreover, we see that they decrease with the size of the seed set, as well as the observation that adding a random seed provides worst marginal gains ( in monotonicity plots) than adding the chosen seed.

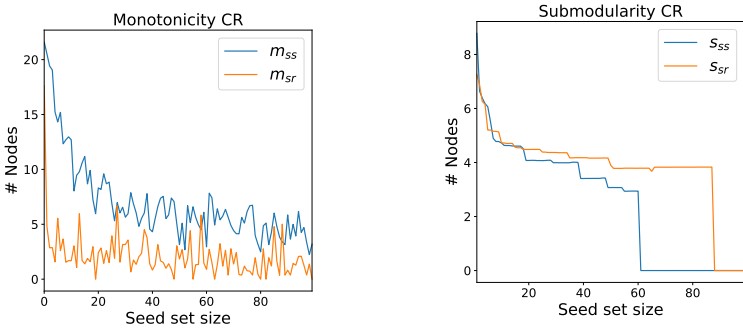

Figure 4: Monotonicity and submodularity for the Crime (CR) dataset.

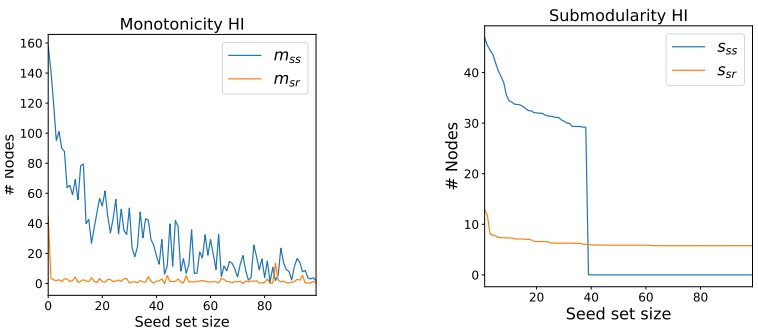

Figure 5: Monotonicity and submodularity for the HI-II-14 (HI) dataset.

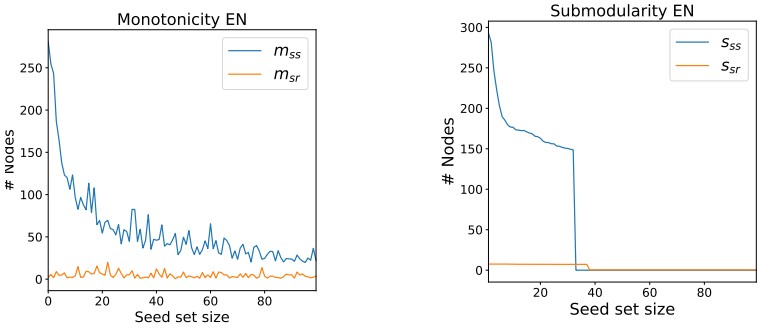

Figure 6: Monotonicity and submodularity for the Enron (EN) dataset.

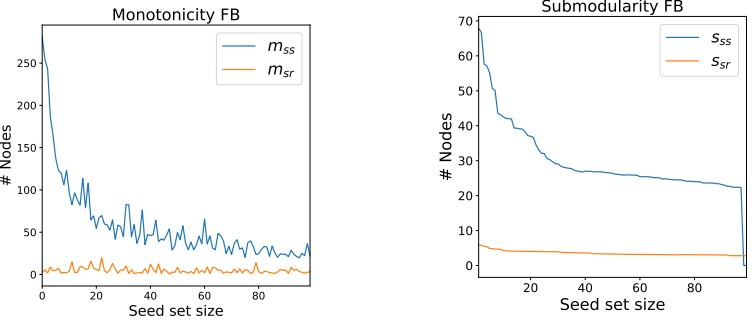

Figure 7: Monotonicity and submodularity for the Facebook (FB) dataset.

## B.4 IMM AND PUN WITH UNIFORM PROBABILITIES

We compare IMM and PUN on the same graphs with uniform influence probabilities $p = 0.01$ in Figure 8. We clearly observe that PUN outperforms IMM.

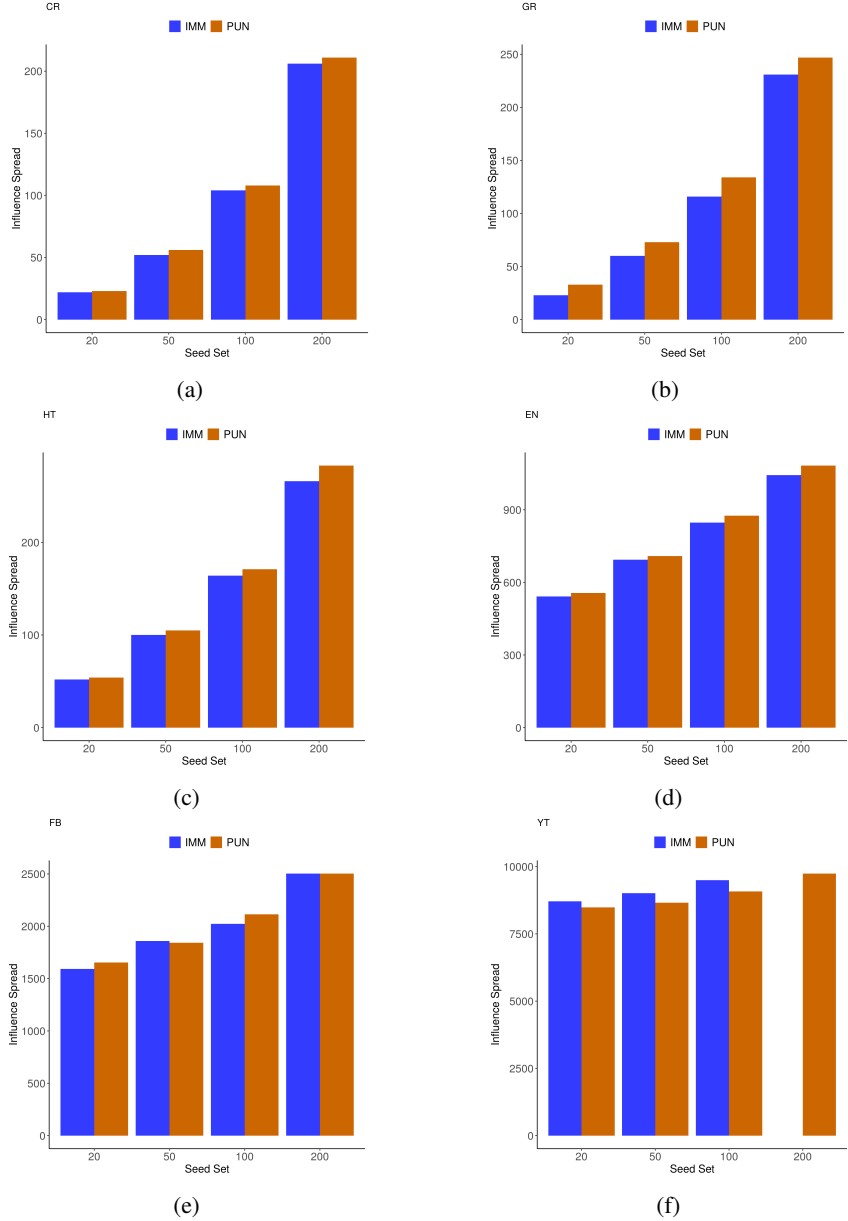

Figure 8: PUN vs. IMM for IC with $p = 0.01$.

# APPENDIX C

## C.5 RESULTS FOR SMALLER SEED SETS

| Graph | GLIE-CELF | GRIM | PUN | K-CORE | PMIA | DEGDISC | IMM | FINDER |
|-------|-----------|------|-----|--------|------|---------|-----|--------|
| CR | 228 | **232** | 227 | 126 | 224 | 227 | 228 | **232** |
| GR | **402** | 208 | 374 | 184 | 342 | 272 | 387 | 379 |
| HI | 1,300 | 1,304 | 1,307 | 866 | 1,274 | 1,234 | **1,316** | 1,110 |
| EN | 8,189 | 8,185 | 8,205 | 5741 | 8,041 | 7,988 | **8,208** | 3,964 |
| FB | **4,487** | 4,480 | 4,263 | 1594 | 2491 | 4,489 | 4,481 | 1,823 |
| YT | **105,897** | 104,824 | 104,106 | 67,084 | 97,622 | 104,706 | 105,888 | 3,799 |

Table 7: Influence spread computed by 10,000 MC ICs for 20.

| Graph | GLIE-CELF | GRIM | PUN | K-CORE | PMIA | DEGDISC | IMM | FINDER |
|-------|-----------|------|-----|--------|------|---------|-----|--------|
| CR | **381** | 371 | 378 | 263 | 378 | 368 | 375 | 367 |
| GR | **738** | 553 | 700 | 303 | 654 | 556 | 725 | 635 |
| HI | **1,914** | 1,905 | 1,908 | 1,407 | 1,899 | 1,824 | 1,589 | 1,904 |
| EN | **11,819** | 11,114 | 11,757 | 9,796 | 11,686 | 11,183 | 10,698 | 7,133 |
| FB | 6,631 | 5,879 | 6,329 | 2,974 | 6,574 | 4,489 | **6,724** | 5,649 |
| YT | 147,631 | 148,250 | 145,796 | 78,575 | 145,863 | 143,161 | **148,597** | 9,152 |

Table 8: Influence spread computed by 10,000 MC ICs for 50.

## C.6 COMPUTATIONAL TIME OF HEURISTICS

| | 100 **seeds** | | | | 200 **seeds** | | | |
|-------|------|---------|--------|-----|------|---------|--------|-----|
| **Graph** | PMIA | DEGDISC | K-CORE | PUN | PMIA | DEGDISC | K-CORE | PUN |
| CR | 0.13 | **0.04** | **0.04** | 0.15 | 0.21 | 0.06 | **0.04** | 0.25 |
| GR | 0.70 | **0.12** | 1.5 | 0.17 | 0.80 | **0.13** | 1.5 | 0.26 |
| HI | 1.24 | 0.13 | **0.12** | 0.17 | 1.36 | 0.14 | **0.12** | 0.27 |
| EN | 24.83 | 1.96 | 2.17 | **0.52** | 26.74 | 2.06 | 2.17 | **0.97** |
| FB | 21.2 | 8.86 | 10.62 | **1.42** | 22.77 | 9.29 | 10.62 | **3.1** |
| YT | 3838.5 | 52.39 | 74.91 | **13.2** | 4006.29 | 54.38 | 74.91 | **28.92** |

Table 9: Computational time of heuristic approaches compared to PUN.

## C.7 PUN'S COMPUTATIONAL TIME WITHOUT GPU

We performed experiments to compare PUN without the use of GPU in Table (10), where it is visible that GPU provides a substantial acceleration, but PUN remains the faster option even without it.

| **Graph** | PUN GPU | PUN CPU | IMM |
|-----------|---------|---------|-----|
| CR | 0.15 | 0.17 | 0.13 |
| GR | 0.17 | 0.27 | 0.57 |
| HT | 0.17 | 0.20 | 0.56 |
| EN | 0.52 | 2.44 | 4.78 |
| FB | 1.42 | 17.5 | 69.9 |
| YT | 13.2 | 97.5 | 55.4 |

Table 10: Comparison between PUN CPU and GPU computational times for 100 seeds.

## C.8 RELATIVE ERROR OF GLIE FOR LARGER SEED SETS

To quantify the potential of GLIE for larger seed sets, we sample 9 random seed sets and 1 with the highest degree nodes and compute the error of DMP and GLIE, with the ground truth influence

divided by the average influence in Table (11). We see that the error does not increase significantly as the seed set increases, and that GLIE outperforms DMP in GR while the reverse happens in CR and HT.

| Graph | Seeds | DMP | GLIE |
|:-----:|:-----:|:----:|:----:|
| CR | 20 | 0.005 | 0.031 |
| CR | 50 | 0.006 | 0.059 |
| CR | 100 | 0.017 | 0.152 |
| GR | 20 | 0.161 | 0.029 |
| GR | 50 | 0.125 | 0.042 |
| GR | 100 | 0.093 | 0.082 |
| HT | 20 | 0.010 | 0.105 |
| HT | 50 | 0.004 | 0.062 |
| HT | 100 | 0.002 | 0.113 |

Table 11: Relative error for diffusion prediction of larger seed sets.

