# OpenReview forum: "Learning Graph Representations for Influence Maximization"
_ICLR.cc/2022/Conference — ICLR 2022 Submitted_

### Official Review · Reviewer_GuP1 · 2021-10-25

**Correctness:** 3
**Technical Novelty And Significance:** 3
**Empirical Novelty And Significance:** 2
**Recommendation:** 6
**Confidence:** 2

**Details Of Ethics Concerns:**

My main concern is with the potential negative applications of influence maximization, which for example could be used to spread false information. Authors indeed explicitly acknowledge these negative aspects as well. However, given that this is a research paper and the fact that there is already abundant work on influence maximization in the literature, I find it unfair to reject the paper based on this ground.

**Main Review:**

The paper studies an important problem data mining problem, i.e., Influence Maximization. The proposed network for approximating the influence is interesting on its own. The paper is overall well-written as easy to follow. However, the paper can benefit from polishing: for instance, the braces in Eq. (5) are a bit confusing. Also, the experimental section is rushed. For example, in Sec. 4.1. it is not clear to me how the graph neural network is trained. Is it trained only on BA graphs? If so, how did the authors use that network for other graphs, when the network weights are clearly dependant on graph node size? Also, it seems that the main novelty of the paper is in proposing to estimate the influence via graph neural networks and the proposed influence maximization methods use already existing techniques in combination with this influence estimation method. In that sense, the theoretical contributions of the paper are marginal. However, combining these techniques is still non-trivial.

**Summary Of The Paper:**

The paper proposes a neural network approach (GLIE) for estimating the influence of a given seed in a given graph. More importantly, the authors propose three different methods to use the proposed Influence Estimation method for Influence Maximization. The authors show the superior performance of their proposed method in comparison with baselines and

**Summary Of The Review:**

Overall, the paper proposes novel methods for Influence Estimation and Influence Maximization. In my opinion, the main weakness of the paper is the missing details in the experimental section.

---

> ### Author Response · Authors · 2021-11-20
> **Answer to Reviewer GuP1, expanded the description of training procedure**
>
> Thank you for your useful evaluation and for focusing on some commonly overlooked, but very important, technical aspects of the work. We try to answer each of your comments separately:
>
> 1. The braces in Eq. (5) indicate concatenation and the equation represents skip connections that are common in GNNs [1].
>
> 2. Thank you for raising this valid concern. We had to keep the training description at a minimum due to the page limit, but after your comment, we have updated that part of the experimental section in **App. B** to provide more intuition as to why we build the training set as such, along with further details about the training.
>    The GNN has indeed trained only in **small simulated BA** graphs. The NN weights of a GNN do not depend on the size of the graph, but solely on the feature dimension $d$ [2], meaning each node is treated as a sample and not as a feature. The features are defined as $X \in R^{n \times d}$ and these are the input to the NN. Note that the product with the Adjacency in eq. (4) does not interfere with the dimensions (it is an $n \times n$ matrix). Hence generalizing to larger graphs is like generalizing to larger test sets in typical ML.
>
>
> 3. Thank you for letting us clarify this. CELF and GRIM are indeed not significantly novel, though their combination with GLIE improves them significantly, but PUN is. It is based on NNs while resembling traditional IM methods and outperforms the rest in terms of combined accuracy and efficiency. We have also included proofs that support its guarantee to a near-optimum solution which is novel among learning-based methods.
>
>
> [1] You, J., Ying, R., & Leskovec, J. (2020). Design space for graph neural networks. arXiv preprint arXiv:2011.08843.
>
> [2] Kipf, T. N., & Welling, M. (2016). Semi-supervised classification with graph convolutional networks. arXiv preprint arXiv:1609.02907.

---

> > ### Comment · Reviewer_GuP1 · 2021-11-23
> > **Thank you for the response!**
> >
> > I thank the authors for their response. They address my question regarding the training on GNN networks. I keep my score.

---

### Official Review · Reviewer_F6tB · 2021-10-30

**Correctness:** 4
**Technical Novelty And Significance:** 3
**Empirical Novelty And Significance:** 3
**Recommendation:** 5
**Confidence:** 3

**Main Review:**

While the use of GNNs for combinatorial optimization has become quite popular in recent years, their application to Influence Maximization (IM) is not straightforward, as it is quite different from previous GNN applications. The IM problem has a strong structure (submodularity), and strong non-learning algorithms are already established for the problem. Therefore, it is quite surprising and encouraging to see that GNNs can actually be quite competitive on this problem.

Another strength of the paper is the thoroughness of the approach. The paper proposes three levels of gradual transition from traditional approach (CELF) to completely estimation-based approach (PUN). Compared to simply providing a single approach that works, this gradual approach allows readers to better understand why a simpler approach does not suffice and more techniques need to be introduced.

One notable weakness of the paper is their lack of reference to Alieva et al https://openreview.net/forum?id=ac288vnG_7U . Since this paper also proposes a learning-based approach for submodular optimization, this approach should've been discussed and compared against as a baseline.

Also, in (7), $O(S, s)$ and (10), authors heuristically construct features from frozen GLIE models. Authors provide some intuition about them, saying the hidden state $H_t^i$ can be considered as "a label for each node whose sign indicates if it is predicted to be influenced." However, this seems like an overstatement: $H_t^i$'s are intermediate hidden states from the GLIE model, and there does not seem to be any mechanism (like a loss function) which encourages these intermediate hidden states to be the "influence label". Many design choices like why (7) sums over $T$ and normalize by $d_t$ are unclear. (10) is even more confusing because it only uses the first layer of the GNN- does it mean that additional GNN layers are not needed? Actually, it doesn't seem like authors share an analysis of the impact of GNN layers.

Lastly, it wasn't sure $\sigma^m(S)$ can be interpreted as a submodular function in a standard sense. For a submodular function, we should be able to evaluate for any set in the domain of the function. However, definition (12) assumes the given input $S$ can be associated with the sequence of seed sets $S_1, S_2, \ldots, S_{|S|}$. In other words, the function is only well-defined for the sequence of sets $S_1, S_2, \ldots$ observed during the execution of the greedy algorithm. Therefore, I am not sure (1-1/e) approximation guarantee of greedy monotone submodular maximization applies here.

**Summary Of The Paper:**

Authors propose several neural network model-based approaches to influence maximization (IM). First, GLIE estimates influence using a GNN, which can be plugged into optimization algorithms like CELF. Second, GRIM avoids the cost of having to estimate the influence of every candidate by approximating the marginal gain with a two-layer MLP. Finally, PUN avoids the cost of having to estimate the influence for every seed node by approximating the influence with features from GNN hidden states. Experiments demonstrate that the proposed method can generalize to graphs significantly different from training data. Also, PUN provides solution quality close to the strongest baseline with a fraction of compute time.

**Summary Of The Review:**

Influence Maximization is a quite surprising application of GNNs. Authors employ a systematic approach to build multiple methods with more departure to the traditional approach and increasing levels of computational advantage. Some of modeling decisions look heuristic, however, and at least requires a better explanation.

---

> ### Author Response · Authors · 2021-11-22
> **Answer to Rev F6tB, update on related work, methodology and proofs**
>
> Thank you very much for your careful and thorough review. Your insightful comments paved the way to improve the paper in numerous crucial parts. We updated our related work to include the paper you cited, extended the methodology to clarify the intuition for our choices on the influenced sets and updated the proofs of submodularity and monotonicity to address the issue you underlined.
>
> 1. Thank you very much for pointing out this very interesting work. We refer to it and contrast it with our paper in the related work. LeaSuRe proposes a clever loss function to learn a policy that approximates submodular greedy optimization in different types of problems. In spite of the obvious overall utility of the method, we have to underline the important hinders on using it for IM.
> The biggest obstacle in such learning for IM is that the proposed $g^{exp}$ training algorithm would require hundreds of hours (or more depending on $N,T$) to run for a $\sigma$ computed by 10k MC ICs on regular or even small graphs.
>
>    We would thus need to pair LeaSuRe with GLIE for fast $\sigma$ and end up with a method somewhat similar to GRIM, as our reward in (9) resembles LeaSure's $g^{exp}$. The main difference being that GRIM is trained using e-greedy while LeaSuRe uses Dagger's method. In this case, we would need a novel neural model $g$ that can capture the IM problem state the same way LeaSuRe uses Deepset. This would constitute a contribution on its own, as such a neural model is not invented and common GNN encoders were proven very ineffective in our initial experiments for GRIM. Moreover, PUN optimizes greedily because of the properties of Glie, which in turn relies on the theory of IM. Learning an approximation policy with an arbitrary neural network encoding would be quite far from traditional IM methodologies, which is the conceptual gap our paper tries to cover.
>
>    Finally, we need to underline the technical difficulties to compare experimentally: First we have to train for IM which requires a novel setup and the novel aforementioned neural model for $g$.  Second, we could not find the code in the openreview supplementary material, in github or by contacting one of the authors, thus we would have to develop the method from scratch. Finally evaluating with 10k MC ICs takes days for some of our networks, which (plus the above) would exceed the rebuttal period.
>
>
> 2.
>    We agree with your point that the explanation is insufficient. We have adjusted the text in **Section 3.1** to clarify the connection between the upper bound and the activations from the learnt parameters.
>
>    To be specific, the derived representations, $H_t^i$, are connected to the theoretical upper bound of the influence probability on the t-hop neighbors of the initial seed set. That connection is justified because of the binary input $X$ that signifies the seed set and the inherent message passing of the GNN with that specific transition matrix. It is established through Eq. (3) and Corollary 3.1. and described in the paragraphs in between.
>
>    Although indeed there is no direct cost function for the intermediate layers, the output in Eq (5) shows that each layer is concatenated and it is used to predict the final number of influenced nodes. Given that each layer starts with an upper bound, we argue that the parameters reduce the influence probabilities produced in each layer $H_t$, such that the final concatenation of $[H_t \forall t]$ can provide an accurate prediction of the total influenced nodes. Hence, it is meaningful to assume $\sum_{i=0}^{d_t} H_t^i$ is a vector connected to the probability of influence, and we observed empirically that   averaging the representation and then summing over layers and thresholding on 0 gives a set of length close to the real influence spread. A heatmap of the layers can be seen in Fig. 1. We use the average because $d_t$ varies throughout layers and we need an equal contribution from each dimension of each layer to the final output for each node, because the sign of the sum determines whether the node is influenced or not. Note that it is not as accurate as the final output of Glie, but it allows us to find **which** and not only **how many** nodes are influenced. The definition of the influenced set for PUN follows the same reasoning.
>
>
> 3. Thank you very much for your keen observation. Indeed it was a mistake to define the influence spread as such.
>
>    We thus **redefined the influence spread in Eq 11** and created new, more elaborate proofs for submodularity and monotonicity of the new function and of the marginal gain approximation in **App A** to retain the theoretical guarantee.
>
>    Note that in practice, this does not affect PUN, which uses directly the marginal gain obtained in $m= A^T\hat{L}$, as shown in Fig. (1), but does not rely on the influence spread. Our new definition is comparable with the same marginal gain, hence we did not have to change the method or rerun the experiments.

---

> > ### Comment · Reviewer_F6tB · 2021-11-22
> > **Thanks for addressing my questions**
> >
> > On 1, I agree that the comparison with LeaSuRe can be nontrivial. I think the including the discussion you provided in the response in either the main body or the appendix would be enough. And thanks for sharing your insights for 2) and making technical improvements for 3).

---

### Official Review · Reviewer_et7w · 2021-10-31

**Correctness:** 3
**Technical Novelty And Significance:** 2
**Empirical Novelty And Significance:** 2
**Recommendation:** 3
**Confidence:** 4

**Main Review:**

I would proceed by saying that I was one of the reviewers of this paper in its previous submission to another conference. First of all, I appreciate the efforts made by the authors to improve the previous version. I am willing to increase my score if the author could address my concerns.

$\textbf{Strength:}$
-	Using GNN for learning representations is a reasonable choice
-	This paper is technically sound.
-	Constructing a submodular score function based on the learned representation is also an interesting idea.

$\textbf{Weakness:}$

Using learning methods for estimating influence or influence maximization have been extensively studied (e.g., [a]-[d] listed below and their references). Using GNN and Q-learning is reasonable but not novel.

[a] Li, Hui, Mengting Xu, Sourav S. Bhowmick, Changsheng Sun, Zhongyuan Jiang, and Jiangtao Cui. "DISCO: influence maximization meets network embedding and deep learning." arXiv preprint arXiv:1906.07378 (2019).

[b] Du, Nan, Yingyu Liang, Maria Balcan, and Le Song. "Influence function learning in information diffusion networks." In International Conference on Machine Learning, pp. 2016-2024. PMLR, 2014.

[c] Xia, Wenwen, Yuchen Li, Jun Wu, and Shenghong Li. "DeepIS: Susceptibility Estimation on Social Networks." In Proceedings of the 14th ACM International Conference on Web Search and Data Mining, pp. 761-769. 2021.

[d] Liu, Qi, Biao Xiang, Enhong Chen, Hui Xiong, Fangshuang Tang, and Jeffrey Xu Yu. "Influence maximization over large-scale social networks: A bounded linear approach." In Proceedings of the 23rd ACM International Conference on Conference on Information and Knowledge Management, pp. 171-180. 2014.

The experiments do not demonstrate a strong need for learning-based methods for influence estimation or influence maximization.

- The influence estimation problem is only challenging when the cascade can spread for a large number of hops. The presented experiments consider influence estimation from small seed sets. Under such settings, even MC is not time-consuming. For the same reason, the independent cascade model (with uniform probability) is a better model to study, as it in general results in a much larger influence than the WC model – for dense graphs, the WC model tends to give low propagation probability.

- Notice that IMM might be the state-of-the-art method with approximation guarantees, but efficient heuristics are possible (as discussed in [e]). It is well-known that heuristic methods can provide solutions with qualities similar to the (1-1/e)-approximation. Therefore, to demonstrate the efficiency of the proposed method, it is better to also compare it with fast heuristics.

[e] Debunking the myths of influence maximization: An in-depth benchmarking study

- The testing size appears to be too small to draw results that are statistically significant. In addition, there is no information about std to demonstrate the robustness.

The proposed learning model essentially uses a diffusion process of bounded steps to estimate the entire diffusion process. Since the layer number is 2, this means that only two steps of diffusion are considered. Under the WC model, it is likely that the majority of the nodes are activated within two steps, and therefore it is not very surprising that the estimate is accurate. Notice that the #p-hardness only holds for general diffusion steps, and assuming that only two diffusion steps are considered, many existing methods can be modified to be more efficient (e.g., IMM with two-hop reverse sampling).

Minor questions:
- What does negative sample mean in Sec4.1?

- It is quite surprising that PUN can produce high-quality solutions to large graphs with a very small computation cost. This essentially means that the #p-hardness could be overcome by a one-step influence estimation, which looks too ambitious. If this is the case, $\sigma^m$ should be a nice method for influence estimation, which could be verified by experiments. In a related issue, for the running time of the proposed methods, I am wondering if the time used for computing the representations is included. For large graphs, computing the representation involves multiplications of large matrices.


**Summary Of The Paper:**

This paper considers using learning methods to solve the well-known influence maximization problem. The paper proposes to estimate the upper bound of the influence by using graph neural networks, which can be used in subsequent steps for selecting the seed nodes through either Q-learning or a greedy algorithm based on the learned representation. Experiments on various datasets have been provided to evaluate the accuracy of influence estimation as well as the effect of influence maximization.

**Summary Of The Review:**

The proposed technique is not very novel, and its practical utility needs better justification.

---

> ### Author Response · Authors · 2021-11-17
> **Answer to Reviewer et7w points, experiments with heuristics, uniform probabilities and larger seed sets**
>
> We are very thankful for your review. Your justifiable concerns have allowed us to clarify the method's effectiveness and the novel parts of the paper.
> We first answer the points that require experiments and continue with the rest.
>
> 1. ("Notice that IMM might be..")
>
> Thank you for your invaluable suggestion. We **perform new experiments** with the three most well known IM heuristics for IC : DegreeDiscount, PMIA, kcores (note LDAG and simpath use the LT) and add 50 and 200 seeds to answer to your concern about the model as the seed set increases and the cascade becomes larger.
> We have updated the tables for 100-200 seeds in section 4.2 and added 20-50 in the **App C.5**.
> We see that **PUN outperforms the heuristics** in accuracy, while being either faster or competitive (App C.6). Moreover we see that PUN's advantage over IMM is **more clear as the seed set increases**.
>
> 2. ("The proposed learning model..")
>
> Although we agree that our model should adapt to different settings, we first have to underline that WC is the most widely adopted model in IM  used in (Kempe et al 2003), IMM, LDAG, PMIA and others, for reasons described further below. That said, we **run experiments with uniform p=0.01** (the most common assignment) to address your valid concern. You can see in the **App. B.4** that **PUN is superior to IMM** in this setting. Moreover, the **increased seed set size** in the new experiments in sec 4.2 indicates robustness in longer cascades.
>
> Regarding your conceptual questions,#p-hardness can hold for limited steps depending on the graph's density (e.g. a fully connected graph). RRS sampling for two hops reduces the simulation time but increases the need for more RR sets to cover enough nodes as candidate seeds, hence fewer hops is not a clear computational advantage.
>
>
> 3. ("Using learning methods for..")
>
> Thank you for letting us clarify this. To the best of our knowledge, **only 2 published papers** deal with learning-based IM. GNN+QL has been indeed used in FINDER (our benchmark) and GCOMB (analyzed in sec 2.), but **our main novelty is not GRIM**.
> The use of GLIE for adaptive and provably submodular IM is the intended final outcome, which we deem novel.
> As mentioned in the introduction, we aim for an ML method able to learn contextual info that algorithms cannot, while being more interpretable and with a **guarantee**, in contrast to FINDER and GCOMB.
> Finally, linking the GNN to the theoretical upper bound through corollary 1 is also novel.
> The difference between our paper and the provided references are:
>
> a) DISCO is a similar architecture to FINDER. IE is not addressed and there is no theoretical guarantee. Another problem is that we could not reproduce the results to compare.
>
> b) learns a model from real diffusion cascades e.g. Weibo, which is separate literature. Most networks do not include real diffusions.
>
> c) DeepIS is a GNN for IE published 2021 that is indeed similar to Glie, however, the authors do not compare with SOTA estimators like DMP (only with GNNs), do not utilize the representations for IM, or compare with actual IM algorithms.
>
> d) proposed group-PG with no learning
>
>
> 4. ("The influence estimation problem.." and "The testing size appears..")
>
> WC is the most widely adopted model because uniform probabilities are considered less realistic since:
> 1. each user can influence differently its neighbors.
> 2. there exists network assortativity (Newman et al 2003).
>
> We address your concern on the IM with uniform probabilities above.
> To answer on the use of small sets, we **extended the set size for IE** and show the relative error in **App C.8**, indicating that it does not change significantly as we increase the seed set.
>
> We have to clarify that **our main target is IM**. We address IE to provide an interpretable and theoretically supported learning-based IM.
> GLIE is inherently limited by the layers and the training labels.
> A model trained on 5 seeds and 300 nodes can not generalize to 100s of seeds and 1m edges in a hard problem. Typical NNs for CO test generalization from 50 nodes to 100[1]. We can not create larger labeled samples due to the complexity of IM.
> This is the reason we use adaptive submodular maximization. Adaptivity allows us to utilize a limited IE model to perform IM, as we remove influenced nodes and seeds every 10 steps and restart from 0 seeds and a smaller graph. Note that the method has a guarantee in the adaptive setting as well.
>
>
> Minor:
>
> Negative samples are seed sets that are not optimum, used to train GLIE in order to improve generalization.
>
> $\sigma_m$ diverges more from DMP than GLIE, as we show in the figures of adaptivity, it is thus a worse estimator. However, it does not aim to solve IE, but rather provides an efficient approximation that can be used for IM.
>
> The time to compute representations given the trained model is taken into account in the results.
>
> [1] Bresson, X. et al. (2021). The Transformer Network for the Traveling Salesman Problem

---

> > ### Comment · Reviewer_et7w · 2021-11-22
> > **Response**
> >
> > The additional experiments are appreciated. But I would not think the author has fully addressed my concerns. I understand that it is not possible to consider all existing methods in one paper, but the justification provided by the authors is not warranted.
> >
> > 1 The updated paper claims that PIMA, DegreeDiscount, and K-cores are the most well-known heuristic methods for IC model, which is not true. See the remarks in [e]: "IRIE and EaSyIM owing to their global estimation procedure outperform the other local estimation based heuristics like degree discount [5] and PMIA [4], which thus, are ignored in this study."
> >
> > 2 "DISCO is a similar architecture to FINDER. IE is not addressed and there is no theoretical guarantee. Another problem is that we could not reproduce the results to compare." This is not a good reason for not considering DISCO as a baseline. For scientific reasons, I would report whatever results I have for DISCO, even if the results do not match the original paper. In addition, I would classify the method proposed in this paper as a heuristic rather than a method with theoretical guarantees.
> >
> > 3 "Most networks do not include real diffusions." Being capable of learning a model from real diffusion is definitely a plus but not a minus. Based on the author's response, I am not convinced that that method cannot be considered as a baseline.
> >
> > 4 "DeepIS is a GNN for IE published 2021 that is indeed similar to Glie, however, the authors do not compare with SOTA estimators like DMP (only with GNNs), do not utilize the representations for IM, or compare with actual IM algorithms." This is not a good reason for not considering DeepIS as a baseline. DeepIS is a valid baseline as long as it solves IM or IE.
> >
> > 5 "proposed group-PG with no learning"  This is not a good reason for not considering it as a baseline. Some baselines adopted in the current paper have no learning either.
> >
> > 6  "We first have to underline that WC is the most widely adopted model in IM". The authors have some misunderstandings about the definition of diffusion models: WC is a special setting of IC, but not a separate model. In addition, the weighted cascade model is used more widely because exactly that it is less time-consuming but not because it is more realistic.
> >
> > 7 "Regarding your conceptual questions,#p-hardness can hold for limited steps depending on the graph's density (e.g. a fully connected graph). " My point was that #p-hardness does not hold if we consider only two-step diffusion.
> >
> > 8 "RRS sampling for two hops reduces the simulation time but increases the need for more RR sets to cover enough nodes as candidate seeds, hence fewer hops is not a clear computational advantage." First, it is possible that fewer hops is not an advantage for IMM, but this has to be verified through experiments. Second, fewer hops is clearly an advantage for most of the heuristics.

---

> > > ### Author Response · Authors · 2021-11-22
> > > **Answer to rev et7w**
> > >
> > >
> > > We first have to underline that we are surprised with your response because it comes **only a few hours** before the end of the discussion period. Given that IM requires **tens of hours to evaluate, it is impossible for us to develop and test more benchmarks now**.
> > >
> > > In response to your request to compare with heuristics, we compared with the **oldest and most established heuristics** while underlying the undeniable differences with the provided references. It was not clear that all **7 baselines** that you mention above are required , which with our current **5** make a total of **12 baseline models**.
> > > We are unaware of **any IM paper** that contains so many different methods and it is not a review paper.
> > >
> > > To address each of your comments:
> > >
> > > 1. (with 2, 4, 5): PMIA is actually one of our baselines. We could not find any open source code for Disco and our implementation returned a $\sigma$ orders of magnitude smaller then the ones reported in our experiments. Our code may be wrong, but developing and debugging every unpublished paper for IM in arXiv should be considered impossible. IRIE, EasyIM, DeepIS and group-PG  are all indeed IM methods that we can compare against. However, this list could include **at least 50 more** methods on IM.
> > > That does not alleviate their **substantial methodological differences** and the possible accompanying advantages, e.g. learning context-based IM, nor that **they do not surpass all methods we compare with**.
> > >
> > >
> > > 3. As mentioned in the response to rev caQs, **learning  diffusion from cascades is a different problem and requires different data i.e. diffusion cascades**. Moreover, it is based on sequential models that can not be used for IM.
> > >
> > > 6. We **did run experiments with a uniform p=0.01 (not WC) and larger seed sets as you suggested**, and they indicated that our **method remains superior**.
> > >
> > > 7. (and 8) We will have to disagree for the reasons mentioned in our initial response.
> > >
> > > To our dismay, **the requested experiments that provide answers to questions (6-8)** by comparing our method in a larger cascade setting were disregarded in your answer.
> > > With regard to the general criticism on the bounded number of steps, these experiments should constitute an answer to the utility of the method. Furthermore we underline that our model **can be extended to multiple steps** based on the new spread devised by the question of Rev. F6tB. We can extract and sum Eq. (11) for multiple layers, and submodularity holds as a sum of submodular functions.
> > >
> > > Finally, regarding the comparisons, our model outperforms a number of methods which constitute general standard baselines for research articles that contribute to the IM problem.
> > > We understand the result-driven ideology, but as mentioned above, we deem 12 benchmarks too much. More importantly, not all requested methods are open sourced and **IM takes hours to evaluate** for just a dataset. Hence a **method may require days in total** for every combination of seed set size and dataset, and the comment coming on 22 Nov (deadline) while our initial response was on 17 Nov, was condemnatory.

---

> > > > ### Comment · Reviewer_et7w · 2021-11-23
> > > > **Response**
> > > >
> > > > The fact is that it is not possible to push this paper above the acceptance threshold within a short response phase. I have made it very clear in my response that I understand that it is not possible to consider all existing methods in one paper, but the point is that the many justifications provided by the authors in the original paper and the response are not warranted.
> > > >
> > > > Your new experiments are appreciated, and they are not disregarded - not changing my score does not mean that they are disregarded. In addition, please be advised that "Area chairs and reviewers reserve the right to ignore changes that are significantly different from the original paper", as noted on the ICLR website. Even with more experiments, I do not think any of my main concerns can be addressed: lacking technical novelty in terms of learning methods, lacking justification on the need of the proposed method, and the fact that the (hidden) idea of limiting the spread of steps for gaining efficiency is not new.
> > > >
> > > > This paper is clearly a good piece of work for second-tier conferences. I do believe that having the existing methods either implemented in the experiment or *appropriately* discussed in the paper could result in a better submission.

---

> > > > > ### Author Response · Authors · 2021-11-23
> > > > > **Answer to the response of rev et7w**
> > > > >
> > > > > Our main argument is that you did not address any of our technical novelties, such as learning the representations to decide **which** nodes are influenced or providing a **learning-based method with a guarantee**, which all past ML for IM methods lack.
> > > > >
> > > > > **"Even with more experiments, I do not think any of my main concerns..."**, meaning that no experiment would change your mind to see the paper as anything but a clear rejection (3), although we:
> > > > >
> > > > > 1. **refuted your arguments for the limited spread** with experimental evidence on different cascade settings.
> > > > >
> > > > > 2. **compared with heuristics** to answer on your initial comment "it is better to also compare it with fast heuristics."
> > > > >
> > > > > while in your first comment you contended
> > > > > **"I am willing to increase my score if the author could address my concerns".**
> > > > >
> > > > > **"for second-tier conferences"** it is clear from your response to **this and the previous conference submission** that you strongly believe our paper does not belong in these conferences and you intend to keep reviewing it in a gatekeeping manner.
> > > > >
> > > > > Having said that, we respect your decision and thank you for asking for further experiments which enhanced the presentation of our method.

---

> > > > > > ### Comment · Reviewer_et7w · 2021-11-23
> > > > > > **Response**
> > > > > >
> > > > > > It was not my opinion (in your last submission) that your paper is a good second-tier conference paper, but it was indeed one of the reasons why **your submission was unanimously rejected by all the reviewers in your last submission**.
> > > > > >
> > > > > > I wish you the best for your this and future submission.

---

### Official Review · Reviewer_caQs · 2021-10-31

**Correctness:** 3
**Technical Novelty And Significance:** 2
**Empirical Novelty And Significance:** 2
**Recommendation:** 3
**Confidence:** 4

**Main Review:**

Strength
1. It is an interesting and practical idea to consider learning-based approach for cascade problems on social networks.
2. The authors carry out experiments on several networks (both synthetic and real-world) with reasonable large number of nodes (>1M).

Weakness
1. Some of the writing in the paper is not clear or misleading. First, it is not clear what is the input to the GNN. It mentioned that H_0 \in R ^{n*d}. If just whether the node is in seed set is encoded, only 1 dimension is needed. Second, it would be better for the authors to clarify that the monotonicity and submodularity of only (12) is proved. There is no guarantee for (6) but only qualitative evaluation for (6). As a result, the usage of CELF may not be well justified.
2. The method with best performance PUN has little to do with the learning method. It is just a diffusion function with 1 step. Also, the network is actually not trained for the target. One possible reason is that the activation probabilities are really small (1/ avg degree) setting such that the cascade depth are really shallow. In this case, the authors should compare their method to some heuristics.
3. The empirical evaluation is not very convincing. First, RSS-based methods are not included in the influence estimation part. Second, running time comparison on GPU based method and CPU method is not a fair comparison. Third, as mentioned above, heuristic-based method should be included in the comparison as in Table 5.
4. One strength of learning-based approach is to generalize beyond one given diffusion model. It would be interesting to see (1) experiment on real-world cascades; (2) generalization under diffusion model misspecification.


**Summary Of The Paper:**

In this paper, the authors consider using learning-based method for Influence Estimation and Influence Maximization. The author proposed a GNN-based to estimate influence (as an upper bound). Based on the estimated influence, the authors use CELF optimization to find the optimal seed set. To further improve the efficiency, the authors proposed (1) a RL DQN based method and (2) a simplified influence function with only one layer. The author carries out experiments on several synthetic and real-world datasets.

**Summary Of The Review:**

Though the idea of using learning-based approach for IM problem is interesting, the proposed method did not demonstrate advantage over existing methods (the best performing one is more like existing heuristics). Also, there are a few missing pieces in the evaluation.

---

> ### Author Response · Authors · 2021-11-17
> **Answer to Reviewer caQs points, experiments with heuristics, CPU and uniform probabilities**
>
> Thank you for your valuable comments, we tried to address your concerns one by one and extended the experimental part of the paper that showcases the empirical strength of our method.
>
> 1.
> a) Thank you for underlining this ambiguous point. The input is multidimensional in order to be usable by the GNN, meaning using solely the binary vector, as you mention, we compute the upper bound. Using high dimensional embeddings of ones is a common practice in GNN literature to learn different graph properties [1]. In a similar manner for our case, higher dimensions allow for combining the features into new representations that benefit the prediction and aim to capture the definite probability of nodes being influenced.
>  b) Although we have not explicitly stated that the theoretical guarantee applies for GLIE-CELF, we have updated our text (sec 3.2) to clarify that this is the case to refrain from further confusion.
> We understand your doubts on the motivation, but apart from the clear pattern in Appendix B.3, the result of GLIE CELF producing SOTA seed sets throughout the overwhelming majority of the experiments should overall justify its use.
>
>
> 2.
>  Thank you for letting us clarify this rather crucial part of our methodology. PUN uses GLie to decide who is predicted to be influenced and computes the proxy for the marginal gain based on that and the adjacency matrix. It thus relies heavily on the learning part, which indicates which nodes are influenced or not. As a direct response to your argument, we **run experiments with the most well-known heuristics** for IC: DegreeDiscount, PMIA, kcores and add it in **Sec 4.2**. The **superiority of PUN over the heuristic** methods is visible.
>    Regarding your comments on WC, please keep in mind that it is the most widely adopted model in IM  used in (Kempe et al 2003), IMM, DegreeDiscount, PMIA etc. That said, we performed experiments to compare PUN with IMM using the **uniform p=0.01** (the most common assignment) instead of (1/ avg degree) to address your concern. We see that **PUN clearly outperforms IMM** in **App. B.4**. YT is not included because IMM and its evaluation have not finished yet, but we will update the paper as soon as it finishes.
>    Finally, as a general addition to the motivation, we would like to highlight that PUN is a learning-based method that has a **theoretical guarantee**, which separates it from the GNN-DQL approaches. Moreover, non-learning heuristics and algorithms can not condition on contextual information as mentioned in the text, which is a clear advantage of learning-based methods that should enhance the motivation of using PUN, along with our experimental evidence.
>
>
>
> 3.
>   a) Thank you for your valid point. We do not use reverse reachable set estimation because it requires the creation of 10k RR sets to perform one IE, which would render it very costly in the context of our experiments.
> The great advantage of RR sets in IM is that they do not require recomputing the sets for each estimation, but in our experiments of individual seed sets the initial overhead would be added in every individual seed set, and hence the computational gain is lost.
>   b) Indeed we should have tested the CPU capacity of PUN. We run experiments without using the GPU  and the results can be seen below.**PUN remains faster than IMM** on the majority of the graphs for **100** seeds (also in **App C.7**):
>   c) Please see the answer to Point 2.
>
> graph | PUN GPU | PUN CPU | IMM
> --- | --- | --- | ---
> CR  | 0.15 | 0.17 | 0.13
> GR  | 0.17 | 0.27 | 0.57
> HT  | 0.17 | 0.20 | 0.56
> EN  | 0.52 | 2.44 | 4.78
> FB  | 1.42 | 17.5 | 69.9
> YT  | 13.2 | 97.5 | 55.4
>
> 4.
> Indeed, a solid method for influence estimation can predict real cascades or work with misspecified models. However, this is the problem of diffusion prediction, where sequential models thrive [2]. These models unfortunately 1. require diffusion to be trained on and 2. can not be used for IM. For the purpose of our study, we argue that training on graphs of 300 nodes and testing on graphs of 5000 nodes is an indicator of generalization that suffice for the purpose of IM, which is the **main target** of the paper. In the literature of GNN for Combinatorial Optimization problems, generalizing from tens to hundreds of nodes is considered succesful[3], and since our model is a GNN inspired by the theoretical upper bound and not a sequential neural architecture trained on diffusions, we contend that the generalization capability is promising and most importantly it suffices for IM.
>
>
> [1] Errica, Federico, et al. "A fair comparison of graph neural networks for graph classification." arXiv preprint arXiv:1912.09893 (2019).
>
> [2] Islam, Mohammad Raihanul, et al. "Deepdiffuse: Predicting the'who'and'when'in cascades." 2018 IEEE International Conference on Data Mining (ICDM). IEEE, 2018.
>
> [3] Kool, Wouter, et al. "Attention, learn to solve routing problems!." arXiv preprint arXiv:1803.08475 (2018).

---

> > ### Comment · Reviewer_caQs · 2021-11-20
> > **response**
> >
> > Thanks author for the detailed response. However, many of the concerns are still not address by the response. First, the theoretical guarantee for PUN is just for a one-step diffusion model and the concern I raise is more about cascade depth. For 3, once with CPU fair comparison, the improvement of efficient is much smaller and we have not even counted for the resources for the training part. For 4, I am not quite convinced about the generalization argument. Therefore, I will remain my original score and think the paper need more work before publication.

---

> > > ### Author Response · Authors · 2021-11-22
> > > **Answer to Reviewer caQs**
> > >
> > > We would like to thank the reviewer for his/her quick response but we would anticipate a more in-depth justification as regards his/her final decision. We did our best in order for the current version of our manuscript to consider all the reviewer's suggestions. Therefore, we would have expected from the reviewer to have a closer look at the revised manuscript. Next, we highlight once again the main findings of the extended empirical analysis suggested by the reviewer.
> > >
> > >
> > > 1. As you suggested, we used a **non-WC** model and **larger seed sets**  that produces different cascade length. The **new experiments** show that we still **outperform** IMM.
> > > We also perform the heuristic comparisons **you proposed** and show the superiority of the model.
> > > Finally and most importantly, with the **new definition** of $\sigma^m$ in Eq. 11 devised to answer comments of Rev. F6tB, **we can actually prove submodularity for more layers** as a sum of submodular functions i.e., $\sigma^m = |Z_S^1| + |Z_S^2|$ for layers 1, 2, etc. However, in practice, we observed this combination to be less successful and efficient. Thus, the argument about the **lack of guarantee for more diffusion steps is not valid anymore**.
> > >
> > >
> > > 2. We run the experiments that you suggested and PUN **remains 2-3 times faster** in the majority of the datasets while providing **better quality** seed sets (as seen from the new experiments about 100-200 seeds).
> > > Regarding the training resources, we assume you refer to the memory consumption, which is larger for IMM due to the need to keep RR sets in memory.
> > >
> > > 3.
> > > As mentioned above, testing to real cascades pertains to different literature that is based on sequential models that **can not be used for IM**.  In the literature of GNNs for combinatorial optimization, Glie's generalization to one scale of magnitude larger networks is considered a success, and in terms of IM, we have a range from 800 to >1 million nodes. Please let us know what would convince you.
> > >
> > >
> > > We kindly request for the reviewer to reconsider her/his evaluation, taking the revision into account.

---

### Author Response · Authors · 2021-11-22
**Summary of revision**

## Summary of changes in rebuttal revision

We thank the reviewers for their detailed evaluation and useful suggestions. We aspire to have answered all the concerns raised through the major revision of the paper.
Below we summarize the biggest changes included in our revision, by section.

[1. Introduction]
We clarified our practical and conceptual motivation for PUN, as well as what separates it from the recent literature on similar problems.

[2. Related work]
We added "Learning to Make Decisions via Submodular Regularization" and clarify the differences with our method, in response to rev F6tB.

[5. Methodology]
In response to the insightful comments by rev. F6tB, we performed two changes in the methodology. We first improved the explanation and intuition behind the use of layer representations to derive influenced set predictions in sec 3.1, arguably the most crucial part of our paper.

Moreover, we redefined PUN's influence spread and adjusted the proofs in sec 3.4 and App A.2 to retain the theoretical guarantee. Note that since PUN optimizes directly for the marginal gain, this had no effect on the method's steps or results.

Finally, we declared that Glie-CELF is not accompanied with theoretical guarantees to refrain from further confusion, in response to rev. caQs.


[6. Experiments]
To address the concern of rev. GuP1, we expanded the explanation and delineate the hyperparameters for the training procedure in Sec. 4 and App B.

In total we perform these additional experiments (shared between sec 4 and App C.) to answer the rightful practical concerns of rev caQs and et7w:
* 3 additional benchmarks to indicate the superiority over heuristics
* Uniform probability diffusion model, for larger cascades
* Larger seed sets from 20-100 to 20-50-100-200, for larger cascades
* Time experiments without the use of a GPU
* Influence estimation for larger seed sets, from 10 to 20-50-100, for larger cascades

---

### Decision · Program_Chairs · 2022-01-20

**Decision:**

Reject

**Comment:**

This paper revisits the problem of influence maximization and suggests using graph neural networks to estimate an upper bound on the influence, which can then be used to find good seed sets. The paper gives a variety of experimental evidence that the methods improve on various algorithms in the literature. There was a wide variation in opinions. Some reviewers felt that the overall idea was not particularly novel, as methods that combine graph embeddings and reinforcement learning to solve influence maximization have already been proposed in the literature. Additionally some reviewers felt that the experiments were missing important comparisons, particularly to learning-based methods, without which it is difficult to argue that these methods really do advance the state of the art.